# Extrusion of subducted crust explains the emplacement of far-travelled ophiolites

Kristóf Porkoláb [1✉], Thibault Duretz[2], Philippe Yamato[2,3], Antoine Auzemery [1] & Ernst Willingshofer[1]

Continental subduction below oceanic plates and associated emplacement of ophiolite sheets remain enigmatic chapters in global plate tectonics. Numerous ophiolite belts on Earth exhibit a far-travelled ophiolite sheet that is separated from its oceanic root by tectonic windows exposing continental crust, which experienced subduction-related high pressure-low temperature metamorphism during obduction. However, the link between continental subduction-exhumation dynamics and far-travelled ophiolite emplacement remains poorly understood. Here we combine data collected from ophiolite belts worldwide with thermo-mechanical simulations of continental subduction dynamics to show the causal link between the extrusion of subducted continental crust and the emplacement of far-travelled ophiolites. Our results reveal that buoyancy-driven extrusion of subducted crust triggers necking and breaking of the overriding oceanic upper plate. The broken-off piece of oceanic lithosphere is then transported on top of the continent along a flat thrust segment and becomes a far-travelled ophiolite sheet separated from its root by the extruded continental crust. Our results indicate that the extrusion of the subducted continental crust and the emplacement of far-travelled ophiolite sheets are inseparable processes.

[1] Faculty of Geoscience, Utrecht University, Utrecht, Netherlands. [2] Univ Rennes, CNRS, Géosciences Rennes, Rennes, France. [3] Institut Universitaire de France (IUF), Paris, France. ✉email: kristof.porkolab@gmail.com

Finding physical mechanisms that explain how dense oceanic lithosphere, referred to as ophiolite, is emplaced on top of lighter continental plates (i.e. obduction) has prompted a long-standing scientific discussion[1–3]. Ophiolites may be accreted to continents by being scraped off from the subducting oceanic lower plate[1,4], or emplaced on top of the continent in a continental lower plate – oceanic upper plate subduction setting[5,6]. In the latter case, subduction typically initiates within the ocean resulting in the formation of high temperature-low/medium pressure (HT-LP/MP) metamorphic soles at the base of the upper plate[7,8]. Intra-oceanic subduction is followed by continental subduction below the oceanic upper plate when the continental margin enters the subduction zone. Such ophiolite belts exhibit high pressure-low temperature (HP-LT) metamorphic units structurally underlying the ophiolite (Fig. 1a–c). These HP-LT units predominantly consist of

continental upper crust that represents the former passive margin of the subducting continent[9–11]. The subduction and exhumation of these units appear to be a relatively short-lived process, as evidenced by the characteristic duration of 10–30 Myr for the subduction-exhumation cycle recorded in the metamorphic rocks (Fig. 1e). The exhumed continental units are typically found between an ophiolite sheet that was transported for tens or even hundreds of kilometers over the continent (far-travelled ophiolite sheets) and the rest of the oceanic domain (open ocean or suture zone of the former ocean; Fig. 1a, b). Far-travelled ophiolite sheets are up to 10 km thick, and their average width is 50–55 km (Fig. 1e), suggesting that their dimensions are mechanically limited during emplacement.

Models of ophiolite emplacement hence have to account for (1) short-lived continental subduction below the oceanic upper plate followed by exhumation of the subducted upper crust, and (2) the

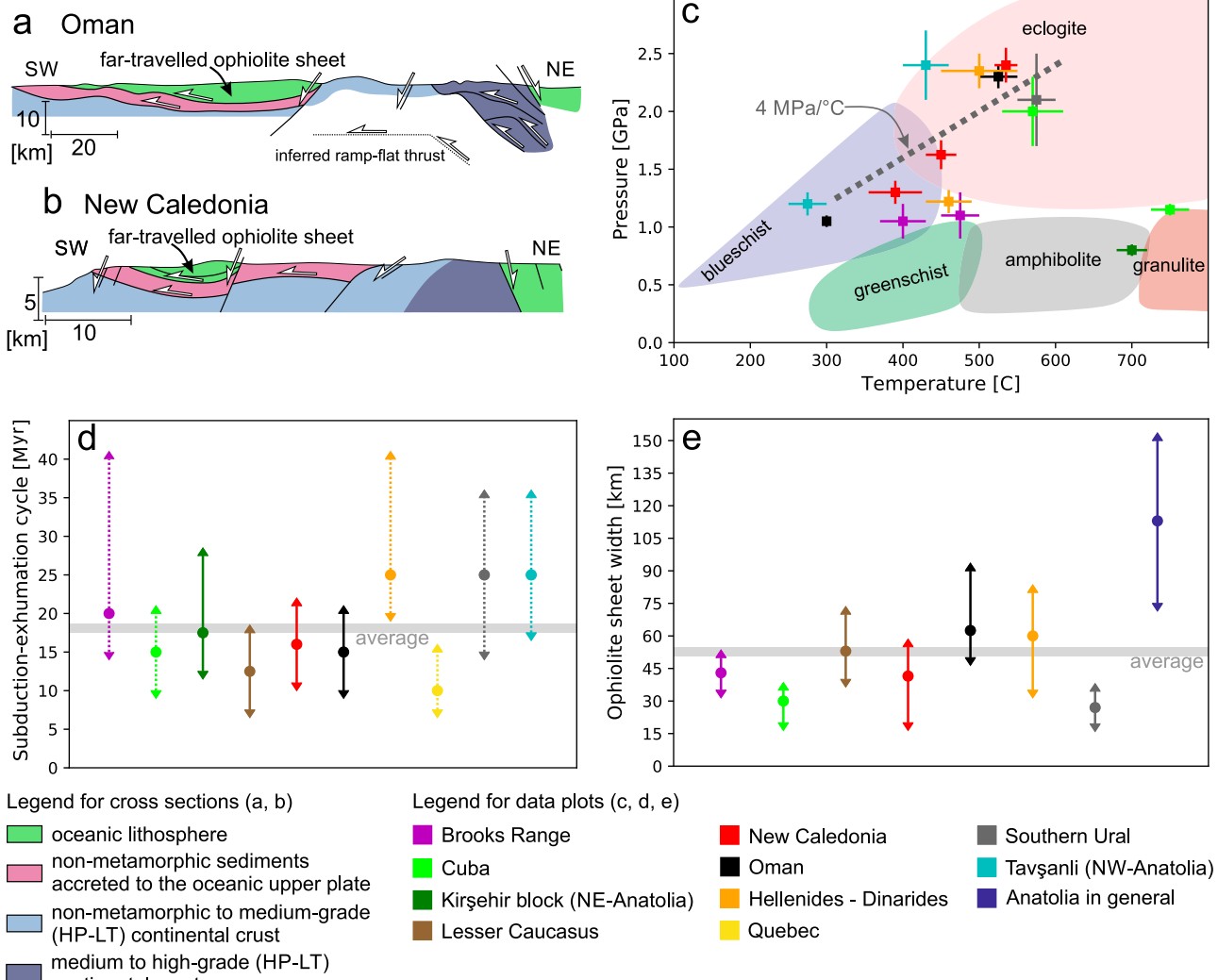

**Fig. 1 Datasets collected from natural ophiolite belts.** For detailed explanation and references regarding the datasets (**c–e**) see Supplementary Note 1 and Supplementary Tables 1 and 2. Simplified cross sections showing the structure of the Oman[13] (**a**) and New Caledonia[56,57] (**b**) ophiolites and their structural relationship to the underlying continental rocks. The far-travelled ophiolite sheets are separated from their oceanic root by exhumed continental units, which show a transition from non-metamorphic to high metamorphic grade. **c** Peak pressure-temperature (*P-T*) data collected from the subducted and exhumed geological units in natural ophiolite belts. Multiple data points from the same location represent the peak conditions of different structural levels. Error bars represent the uncertainty of thermodynamic calculations. **d** Duration of the continental subduction-exhumation cycle below the oceanic upper plate as recorded by the buried and exhumed continental formations. The duration is calculated from the arrival of the continental margin in the subduction zone to the near surface/surface exhumation of the subducted formations and is indicated by colored circles. Error bars show uncertainties associated with the duration of the subduction-exhumation cycle, where solid lines reflect well-constrained and dotted lines poorly constrained cases, respectively (see Supplementary Note 1). **e** Average width of far-travelled ophiolite sheets (colored circles) and the deviation from the average along strike (error bars) in natural ophiolite belts. The width was measured along 3–8 cross sections along each ophiolite belt.

observed dimensions of ophiolite sheets. Recent numerical modelling studies achieved these conditions by imposing convergence to reach the state of continental subduction, and subsequently imposing divergence to exhume the continental margin and thin the oceanic upper plate[6,12]. However, these models do not lead to nappe formation in the subducted continental crust, which is a key feature of natural systems and seems to play an important role in the exhumation of the HP-LT continental rocks[9,13,14]. Moreover, stacking of continental nappes below the oceanic upper plate is critical for inducing uplift on top of the rising thrust sheet(s), which in turn may lead to gravity-driven extension in the upper plate[15–17]. Extension in the upper plate might significantly contribute to the unroofing of the subducted continental rocks as well as in separating the far-travelled ophiolite sheet from its root[15,17]. Hence, exploring decoupling and nappe formation mechanisms in the continental lower plate is crucial for understanding the deformation of the oceanic upper plate and far-travelled ophiolite emplacement.

Here, we combine numerical thermo-mechanical simulations of oceanic upper plate-continental lower plate subduction systems and data acquired in ophiolite belts worldwide to unravel the physical processes that explain the structure of ophiolite belts. We highlight the genetic link between the extrusion of the subducted continental crust and the emplacement of far-travelled ophiolite sheets, and further identify the key parameters controlling this process.

## Results

**Modelling strategy**. We designed 2D thermo-mechanical numerical simulations governed by momentum, mass, and heat conservation equations and a visco-elasto-plastic rheological model (see Methods section for details of numerical modeling techniques and further model setup description). A total plate convergence velocity of 3 cm/yr is achieved by prescribing constant normal inflow velocities of $|V_{in}| = 1.5$ cm/yr along the upper 140 km of the two model sides. Mass conservation is satisfied by gradually increasing outflow below 140 km (Fig. 2a). A convergence velocity test was performed to explore the role of this parameter (Supplementary Note 2 and Supplementary Fig. 1). The top boundary of the model is a true free surface[18]. The initial model geometry is inspired by reconstructions of pre-obduction geodynamic settings where intra-oceanic subduction is initiated relatively close (<400 km) to the continental passive margin, which arrives to the subduction zone after ~10 Myr of oceanic subduction[19–21]. Subduction is designed to initiate along an inclined weak zone (equivalent to an oceanic detachment) close to the mid-ocean ridge to achieve a right-dipping subduction zone and a thermally young ophiolite front, in agreement with natural ophiolite belts[22]. A simplified continental passive margin geometry is implemented by linearly decreasing crustal thickness over the distance of 200 km (Fig. 2a). The continental basement is divided in two parts (upper and lower crust) constituted of different materials (Tables 1 and 2), which introduces a decoupling level at the base of the upper crust (Fig. 2a). A resolution test was performed to verify the robustness of the model results (Supplementary Fig. 2).

**General evolution of the reference model**. Intra-oceanic subduction initiation is followed by the subduction of oceanic lithosphere until 10 Myr. The passive margin of the continent then starts to subduct below the oceanic upper plate (i.e. obduction; Fig. 2b), and experiences HP-LT metamorphism up to eclogite facies P-T conditions (Fig. 2c). The 3-km-thick sedimentary cover of the passive margin partially subducts with the rest of the continental lithosphere, but is largely stacked and

accreted to the front of the oceanic upper plate (Fig. 2). After reaching eclogite facies conditions, the burial velocity of the upper crust decreases from ~3 cm/yr at 18 Myr to near zero at 23 Myr. This is the moment when the upper crust starts to decouple from the lower crust and the lithospheric mantle. Decoupling results in the localization of a major reverse-sense shear zone along which the subducted upper crust is extruded upwards and leftwards (Fig. 2d). The oceanic upper plate undergoes gravity-driven extension when being pushed up by the extruding thrust sheet. Extension leads to the breaking of the oceanic upper plate, which enables the continental upper crust to be rapidly extruded to the surface, separating a 50-km-wide and maximum 13-km-thick far-travelled ophiolite sheet from the rest of the oceanic lithosphere (Fig. 2d, e).

**Ductile nappe formation: the onset of exhumation**. Burial of continental crust below a denser oceanic plate leads to a progressive increase of the buoyancy force that resists subduction despite the continuously imposed plate convergence. From ~21 Myr (i.e. after 11 Myr of continental subduction) the velocity of the subducting continental upper crust becomes near zero, while the lower crust and the lithospheric mantle still subduct at ~1.5 cm/yr (Fig. 3a). This kinematic setting leads to an increase of deviatoric stress within the lower crust, which meets the conditions for strain localization by thermal softening[23]. As a result, a major reverse-sense shear zone develops and exhibits strain rates of $>10^{-13}$ s$^{-1}$ (Fig. 3a). The shear zone is initially low-angle, and propagates along the base of the upper crust, which is the weakest horizon in the continental crust (Figs. 2a and 3a). The shear zone thus facilitates decoupling between the crustal layers, which is further accommodated by distributed deformation (folding) at a strain rate of $~10^{-14}$ s$^{-1}$ in the entire subducted upper crust.

**Interplay between crustal extrusion and upper plate necking**. Upper crustal decoupling and nappe formation results in uplift, which triggers extension of the oceanic upper plate over a 50-km-wide zone (Fig. 3a). Extension leads to the formation of normal faults and to the normal-sense reactivation of the original plate boundary thrust (Fig. 3a).

Further strain localization results in the connection of the two flat reverse-sense shear zone segments by a steeper ramp segment. This structure separates very-low-grade to non-metamorphic upper crust from the low-grade to eclogite-facies upper crust (Fig. 3b). The localization of the main shear zone facilitates the acceleration of upper crustal extrusion (from 0.5–1 cm/yr until 24.5 Myr to 1.5–3 cm/yr from 25 Myr), which is accommodated by increasing displacement along the right-and left-dipping extensional shear zones and leads to the necking of the oceanic upper plate (Fig. 3b, c). Through this process a sheet of oceanic lithosphere (the future far-travelled ophiolite) gets disconnected from the oceanic plate along the left-dipping normal fault, which joins the upper flat segment of the main thrust in depth (Fig. 3b). Subsequently, the far-travelled ophiolite sheet is emplaced on top of the continent and transported to the left along the upper flat thrust segment (Fig. 3c). This process is driven by the progressive extrusion of the continental crust.

**Key parameters controlling crustal extrusion and far-travelled ophiolite emplacement**. Extensive tests have been performed on the reference model to determine key parameters that control the extrusion of the upper crustal thrust sheet, which is instrumental for the necking of the upper plate and the emplacement of far-travelled ophiolite sheets (Fig. 4). In particular, we evaluated the impact of the crustal rheology of the subducting plate and shear heating on the process of nappe formation.

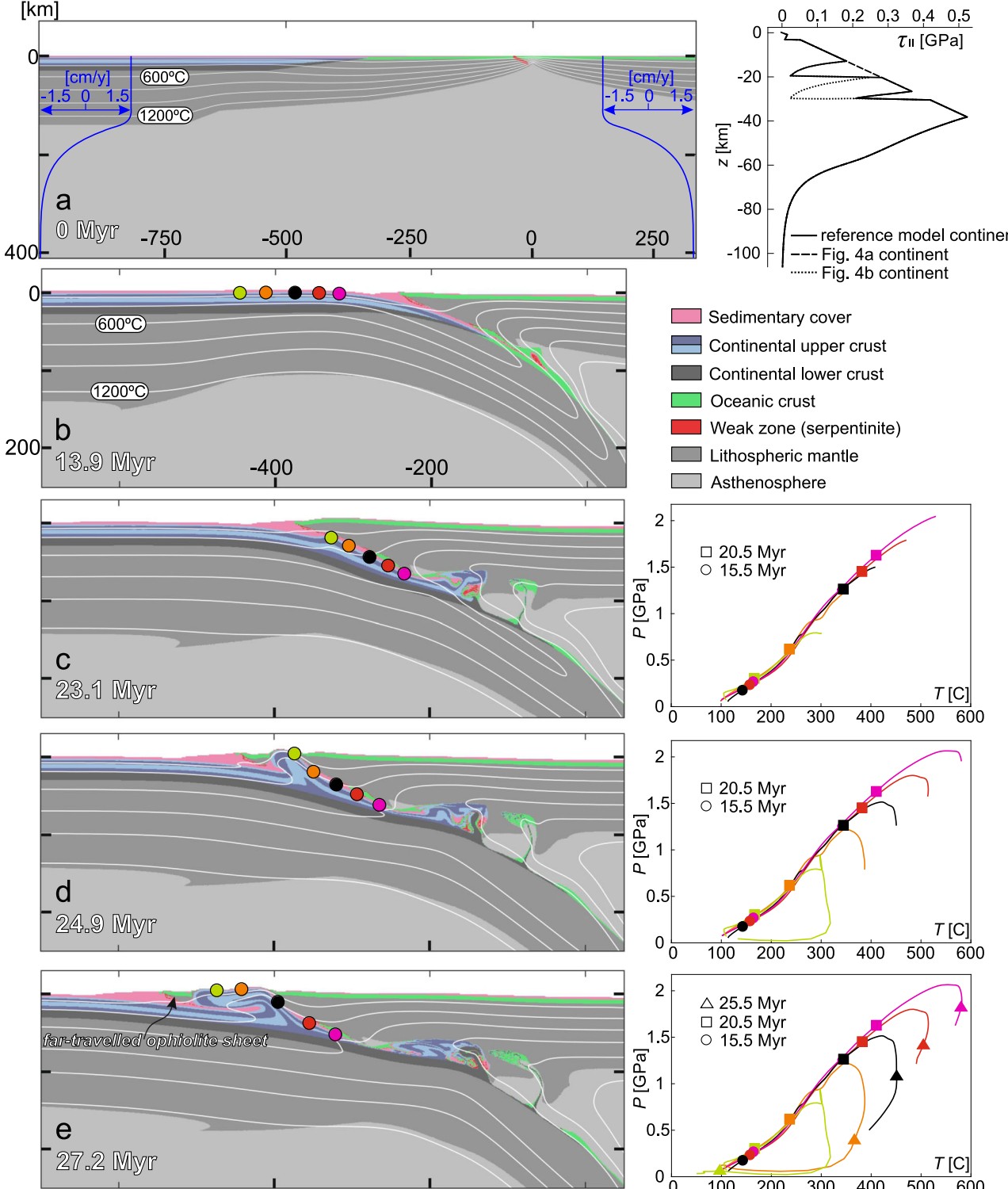

**Fig. 2 Model setup and evolution of the reference model.** The compositional evolution and associated *P-T* paths are presented in **b**–**e** on the left and right side, respectively. Colored circles are the positions of selected particles for recording *P-T* conditions and correspond to the colored lines on the left-side *P-T* diagrams. **a** Model setup showing the initial distribution of compositional domains, the velocity boundary condition, and the strength profile (second stress invariant vs depth, assuming a transcurrent setting) for the continental domain. For thermal and rheological parameters of the model compositions see Table 1. Layering in the continental upper crust corresponds to passive strain markers. White lines are isotherms from 200 to 1200 °C. **b** Early-stage continental subduction at 13.9 Myr after the subduction of the left-side ocean between 0 and 10 Myr. **c** Late-stage continental subduction after the initiation of upper crustal decoupling. **d** Breaking of the oceanic upper plate and accelerated extrusion of the subducted upper crust. **e** Emplacement of the far-travelled ophiolite sheet further leftwards on top of the continent, dictated by crustal extrusion.

**Table 1 Thermal and rheological parameters used for different compositions in the reference model.**

|  | $\rho$ (kg m$^{-3}$) | $k$ (W m$^{-1}$ K$^{-1}$) | $Q_r$ (W m$^{-3}$) | $\alpha$ (K$^{-1}$) | $C$ (MPa) | $\phi$ (°) | $A$ (Pa$^{-n}$ s$^{-1}$) | $n$ | $Q$ (J mol$^{-1}$) |
|---|---|---|---|---|---|---|---|---|---|
| Sedimentary cover (mica) | 2700 | 2.55 | 2.9e-6 | 3.0e-5 | 10 | 15 | 1.0e-138 | 18 | 51.0e3 |
| Continental upper crust (Westerly granite) | 2750 | 2.8 | 1.65e-6 | 3.0e-4 | 10 | 30 | 3.1623e-26 | 3.3 | 186.5e3 |
| Continental lower crust (mafic granulite) | 2900 | 2.8 | 1.65e-6 | 3.0e-4 | 10 | 30 | 8.8334e-22 | 4.2 | 445.0e3 |
| Oceanic crust (Maryland diabase) | 2900 | 3.0 | 1.0e-10 | 3.0e-5 | 10 | 30 | 3.2e-20 | 3.0 | 276.0e3 |
| Lithospheric mantle (dry olivine) | 3300 | 3.0 | 1.0e-10 | 3.0e-5 | 10 | 30 | 1.1e-16 | 3.5 | 530.0e3 |
| Asthenosphere (dry olivine) | 3300 | 3.0 | 1.0e-10 | 3.0e-5 | 10 | 30 | 1.1e-16 | 3.5 | 530.0e3 |
| Weak zone (serpentinite) | 2900 | 3.0 | 1.0e-10 | 3.0e-5 | 0 | 30 | 4.4738e-38 | 3.8 | 8.9e3 |

The heat capacity ($C_p$) and the compressibility ($\beta$) were set to 1050 J kg$^{-1}$ K$^{-1}$ and 10$^{-11}$ Pa$^{-1}$ for all compositions, respectively. Rheological parameters (pre-exponential factor ($A$), stress exponent ($n$), and creep activation energy ($Q$)) are set according to flow laws of mica[59], Westerly granite[24], mafic granulite[25], Maryland diabase[26], dry olivine[60], and serpentinite[61], from top to bottom in the table. Other material properties are $\rho$, $k$, $Q_r$, $\alpha$, $C$, and $\phi$.
$\rho$ density, $k$ thermal conductivity, $Q_r$ radiogenic heat production, $\alpha$ coefficient of thermal expansion, $C$ cohesion, $\phi$ friction angle.

**Table 2 Thermal and rheological parameters used for different compositions in the Mod 1 model variant.**

|  | $\rho$ (kg m$^{-3}$) | $k$ (W m$^{-1}$ K$^{-1}$) | $Q_r$ (W m$^{-3}$) | $\alpha$ (K$^{-1}$) | $C$ (MPa) | $\phi$ (°) | $A$ (Pa$^{-n}$ s$^{-1}$) | $n$ | $Q$ (J mol$^{-1}$) |
|---|---|---|---|---|---|---|---|---|---|
| Sedimentary cover (mica) | 2700 | 2.55 | 2.9e-6 | 3.0e-5 | 10 | 15 | 1.0e-138 | 18 | 51.0e3 |
| Continental upper crust (wet quartzite) | 2750 | 2.8 | 0.5e-6 | 3.0e-4 | 10 | 30 | 5.0717e-18 | 2.3 | 154.5e3 |
| Continental lower crust (felsic granulite) | 2900 | 2.8 | 0.5e-6 | 3.0e-4 | 10 | 30 | 2.0095e-21 | 4.2 | 243.0e3 |
| Oceanic crust (Maryland diabase) | 2900 | 3.0 | 1.0e-10 | 3.0e-5 | 10 | 30 | 3.2e-20 | 3.0 | 276.0e3 |
| Lithospheric mantle (dry olivine) | 3300 | 3.0 | 1.0e-10 | 3.0e-5 | 10 | 30 | 1.1e-16 | 3.5 | 530.0e3 |
| Asthenosphere (dry olivine) | 3300 | 3.0 | 1.0e-10 | 3.0e-5 | 10 | 30 | 1.1e-16 | 3.5 | 530.0e3 |
| Weak zone (serpentinite) | 2900 | 3.0 | 1.0e-10 | 3.0e-5 | 0 | 30 | 4.4738e-38 | 3.8 | 8.9e3 |

The heat capacity ($C_p$) and the compressibility ($\beta$), were set to 1050 J kg$^{-1}$ K$^{-1}$ and 10$^{-11}$ Pa$^{-1}$ for all compositions, respectively. Rheological parameters (pre-exponential factor ($A$), stress exponent ($n$), and creep activation energy ($Q$)) are set according to flow laws of mica[59], wet quartzite[25], felsic granulite[25], Maryland diabase[26], dry olivine[60], and serpentinite[61], from top to bottom in the table. Other material properties are $\rho$, $k$, $Q_r$, $\alpha$, $C$, and $\phi$.
$\rho$ density, $k$ thermal conductivity, $Q_r$ radiogenic heat production, $\alpha$ coefficient of thermal expansion, $C$ cohesion, $\phi$ friction angle.

In the reference model, a decoupled crustal rheology is achieved by using the Westerly granite flow law[24] for the continental upper crust, and a mafic granulite flow law[25] for the lower crust (Fig. 2a). Using a stronger Maryland diabase flow law[26] for the upper crust results in a more coupled, stronger crustal rheology (Fig. 2a). This prevents upper crustal decoupling after reaching eclogite-facies conditions and inhibits subsequent nappe formation and extrusion (Fig. 4a). Instead, the upper crust is subducted to greater depth and enhanced underplating occurs compared to the reference model.

A more coupled, but relatively weak crustal rheology can be achieved by decreasing the strength of the continental lower crust (Figs. 2a and 4b). Such a rheology results in the decoupling of the entire continental crust from the lithospheric mantle rather than the decoupling of the upper crust from the lower crust (Fig. 4b). This leads to distributed folding and thrusting in the lower plate rather than localized nappe formation and extrusion of the subducted upper crust. Our results thus indicate that both strong (Fig. 4a) and weak (Fig. 4b) coupled crustal rheologies inhibit upper crustal extrusion and associated the emplacement of far-travelled ophiolites. We further tested different types of decoupled crustal rheological models to determine the effect of smaller compositional or thermal differences. The results show that slight variations in the thermal and material properties lead to different timing and thus position of crustal decoupling, which is reflected in different amounts of underplated continental upper crust below the oceanic upper plate (Supplementary Fig. 3). Rheology and coupling of the continental crust are hence key factors that control nappe formation, upper crust extrusion, and upper plate necking.

The degree of coupling in the subducting continental crust is also affected by shear heating, a process that dissipates mechanical energy in the form of heat, facilitating shear zone formation in the lithosphere[23,27]. In our reference model shear heating results in heat production that can be up to two orders of magnitude larger than the average radiogenic heat production in the continental crust (Fig. 5b, see Table 1 for radiogenic heat production ($Q_r$) values). This effect is transient and has limited spatial extent, but it induces thermal softening of the ductile material and may trigger the formation of a shear zone. This is documented by a reduction of the effective viscosity at the base of the incipient nappe at the boundary of the lower and upper continental crust and further along the steeper ramp section (Fig. 5). This allows an upper crustal nappe to decouple from the rest of the subducting lithosphere and extrude upwards and leftwards along the shear zone (Figs. 3b and 5). The model which does not include shear heating (Fig. 4c) shows that strain localization and nappe formation fails to initiate, or is significantly delayed compared to the reference model (Fig. 3). Without shear heating, the coupling between the upper crust and the subducting lithosphere is too high to allow for nappe formation and its subsequent extrusion. Instead, buoyancy force leads to the underplating of the subducted upper crust. This demonstrates that shear heating is essential for strain localization and nappe formation in our reference model.

**Comparison to natural ophiolite belts.** Our models display many first-order features of natural ophiolite belts that are related to continental subduction below an oceanic upper plate (Fig. 6). It produces a far-travelled ophiolite sheet as the structurally highest unit, separated from its root by buried and then exhumed HP-LT continental rocks, and underlain by accreted non-metamorphic to low-grade sedimentary cover units (Figs. 1a, b and 6). The tectonic window of exhumed continental crust shows a gradual decrease in metamorphic grade from blueschist-eclogite facies to non-metamorphic towards the far-travelled ophiolite sheet (Fig. 6). The reference model has a relatively warm initial continental geotherm ($T_{MOHO}$ = 510 °C), and therefore shows higher peak metamorphic grades than the Mod 1 variant, which has an initially colder continent ($T_{MOHO}$ = 375 °C) (Fig. 6a, b, respectively).

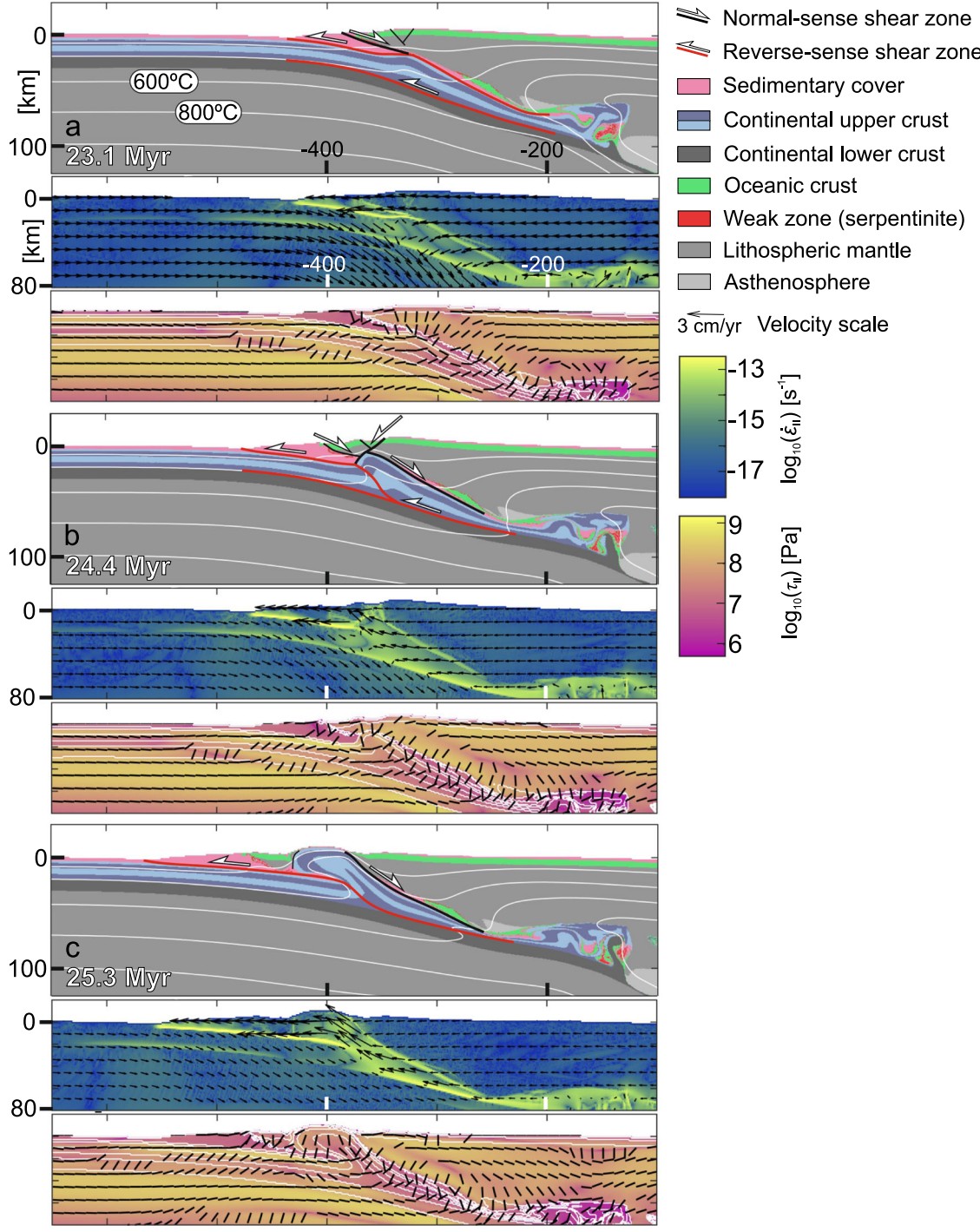

**Fig. 3 Kinematic and dynamic evolution of continental upper crust extrusion and upper plate necking.** Top panels show the compositional domains with the interpreted main structures (shear bands in the brittle domain are interpreted as faults), the middle panels provide information of the strain rate with velocity arrows, while the bottom panels on the second stress invariant with the orientation of the most compressive principal stress ($\sigma_1$). The color fill of the middle and bottom panels are based on scientific colormaps[58]. The orientations of $\sigma_1$ on bottom panels are shown by black lines, horizontal for compression, and vertical for extension. The boundaries of different compositional domains presented on the top panels are highlighted with white contour lines on the lower right panels. **a** Decoupling of the upper crust and initiation of upper plate extension. **b** Further localization of main thrust and extensional accommodation structures that break the upper plate. **c** Emplacement of far-travelled ophiolite sheet by further crustal extrusion.

As such, the reference model closer resembles natural sites where the tectonic window largely consists of metamorphosed crust. Such a case is observed in New Caledonia or Western-Central Anatolia, where the majority of the exposed continental crust reached $P > 1$ GPa and $T$ between 300 and 600 °C[14,28–30]. On the other hand, Mod 1 fits better to natural examples where large fractions of the tectonic window exhibit no or very low-grade metamorphic overprint, with relatively minor volumes of high-$T$ blueschist and eclogite facies rocks (e.g. the Oman or Lesser Caucasus ophiolite belts[12,31]). In both cases, the highest-grade continental rocks are separated from the oceanic lithosphere to the right by a major normal-sense shear zone that accommodated

the extrusion of the subducted continental crust (Fig. 6). The ophiolite sheet and the underlying accreted sedimentary units are resting on the upper flat thrust segment of the main shear zone system that further transported the upper plate units (far-travelled ophiolite sheet and accreted sediments) leftwards during the extrusion of the continental crust. The normal-sense shear zone that separated the far-travelled ophiolite sheet from its root

during the extrusion joins this upper flat thrust segment. The prograde $P$–$T$ ratio and peak $P$–$T$ conditions recorded by the deeper parts of the subducted upper crust in our reference model (4 MPa/°C and ~2 GPa at 500–600 °C, respectively) are average values compared to those of natural cases (Figs. 1c and 2d and Supplementary Table 1). The duration of the continental subduction-exhumation cycle (15–20 Myr) is in agreement with the well-constrained ophiolite belts like Oman or New Caledonia, and appears to be slightly shorter than the sites where subduction and/or exhumation of the continental formations is poorly dated or debated (e.g. Brooks Range, Hellenides-Dinarides, and Southern Ural; see Fig. 1e, Supplementary Note 1, and Supplementary Table 2). Observations such as post-subduction ductile to brittle extensional deformation[13,32], or the coexistence of opposite shear sense directions in the former passive margin units[11,33] fit well with the structural evolution of our model, which involves top-left shearing (thrusting) during burial and both top-left and top-right shearing during exhumation (normal faulting; Figs. 3 and 6). The width of the far-travelled ophiolite sheet (50 km) predicted by the model(s) agrees very well with the average width (53 km) of natural ophiolite belts (Fig. 1e and Supplementary Note 1).

Hence, the structure, dimensions, composition, and $P$-$T$-$t$ conditions of our model closely resemble those of natural ophiolite belts. The geometry and time evolution of our model also allows comparison with the currently active continental subduction of the Australian continental margin below the oceanic Banda arc. The Australian continental margin has been subducting for ~10 Myr[34], which roughly equals the duration of upper crustal subduction in our reference model. Reconstruction of the stacked and accreted sedimentary cover shows that 215–230 km of continental lithosphere has been subducted, but only the uppermost 2 km of sedimentary cover was accreted to the upper plate[35]. Hence the present-day structure and dimensions of the Australian continental subduction are very similar to those of our reference model in the 20–23-Myr snapshots (Fig. 2c).

## Discussion

Our results have important implications for the dynamics of ophiolite emplacement in oceanic upper plate-continental lower plate subduction systems, and thus may be widely applied to understand geological observations in natural ophiolite belts. While plate kinematic changes may play an important role in initiating intra-oceanic subduction[36], or in the cessation of contraction at continental subduction zones[6], we show that the emplacement of far-travelled ophiolite sheets can result from syn-convergence, buoyancy-driven decoupling, and upward extrusion of the subducted continental upper crust accommodated by the necking of the oceanic upper plate. Crustal extrusion and the

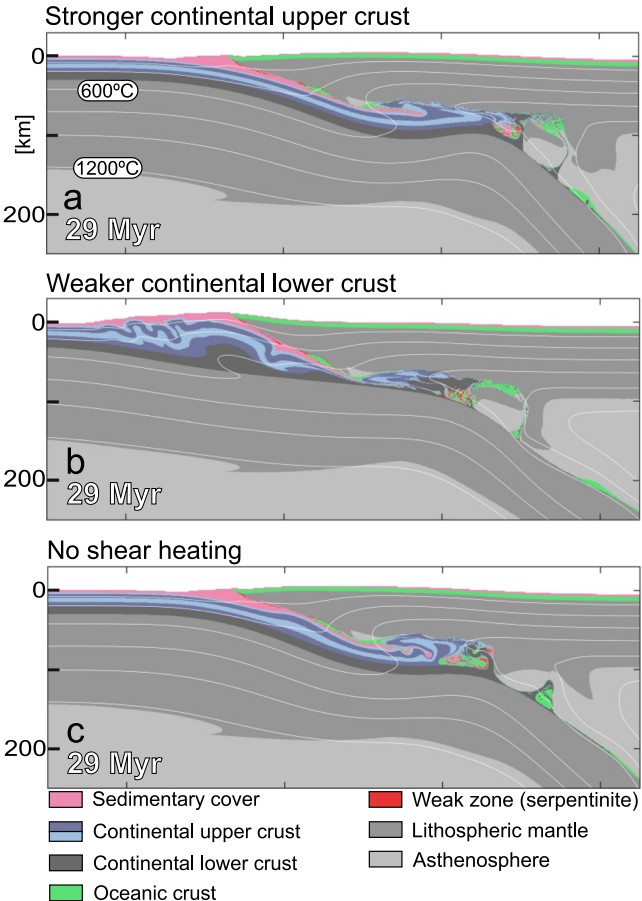

**Fig. 4 Effects of changing key parameters compared to the reference model setup. a** Model compositions at 29 Myr with stronger upper crust rheology (Maryland diabase instead of Westerly granite)[24,26]. For the initial strength profile of the continental domain see Fig. 2a. **b** Model compositions at 29 Myr with weaker lower crust rheology (Maryland diabase instead of mafic granulite)[25,26]. For the initial strength profile of the continental domain see Fig. 2a. **c** Model compositional domains at 29 Myr with shear heating switched off.

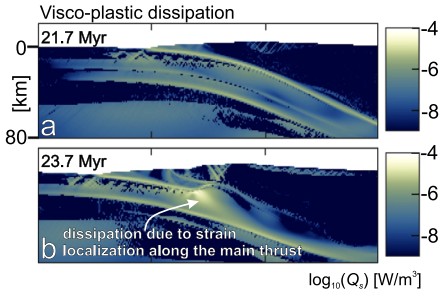

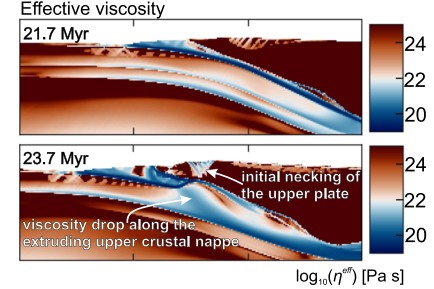

**Fig. 5 The effect of shear heating. a** Plot of visco-plastic dissipation of mechanical energy in the form of heat ($Q_s$; left panel) and effective viscosity ($\eta^{eff}$, right panel) at the 21.6 Myr snapshot of the reference model, prior to nappe formation in the subducting continental crust. **b** Plots of visco-plastic dissipation of mechanical energy in the form of heat (left panel) and effective viscosity (right panel) at the 23.7 Myr snapshot of the reference model, after nappe formation at in the subducted continental crust. Color fills of the plots are based on scientific colormaps[58].

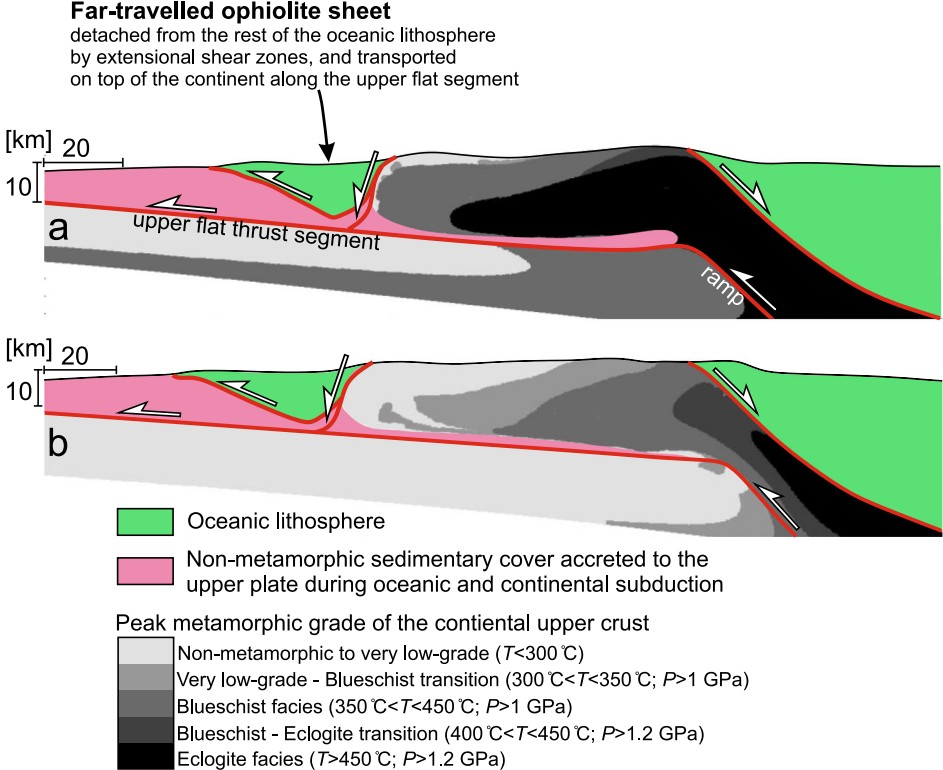

**Fig. 6 Near-surface composition and structure of final model snapshots.** The peak metamorphic zonation of the exhumed continental upper crust is plotted for **a** the reference model and **b** the Mod 1 model variant (for model parameters see Table 2, for initial strength profile see Supplementary Fig. 3). The panels are a combination of calculated metamorphic domains for the continental upper crust, compositional domains for the sedimentary cover and the oceanic lithosphere, and line interpretations of the main shear zones. The peak metamorphic zones are defined based on the recorded maximum pressure (P) and temperature (T) values of the upper crustal particles throughout the entire model evolution, and hence represent the theoretical maximum metamorphic grade reached by the given crustal domains. In both cases, a transition from non-metamorphic to eclogite-grade metamorphic rocks are observed within the upper crustal tectonic window. The reference model (Fig. 6a) has a higher initial continental geothermal gradient (TMOHO = 510 °C), therefore it shows higher peak metamorphic grades compared to the model (Mod 1) with colder Moho temperature (TMOHO = 375 °C; Fig. 6b).

resulting necking of the upper plate in our model are consistent with cases studies from Oman[17] and New Caledonia[15]. Extrusion of the subducted upper crust requires nappe formation. In agreement with previous studies[26,30], the results show that shear heating is an important mechanism that facilitates strain localization and nappe formation. The precise reproduction of smaller scale nappes (nappe thickness of several kilometers) which is often observed in case of continental subduction[9,13] would require very high-resolution numerical modeling and built-in heterogeneities inside the upper crust to localize shear zones at multiple horizons[37–39]. Our results also indicate that crustal decoupling and exhumation may take place with different timing and position in the subducted continent depending on the rheology of the continental crust[40,41]. Variations in thermal or compositional properties thus might control the surface preservation (exhumation) or the subduction and recycling of different types of continental passive margins (e.g. magmatic or magma-poor passive margins)[42–44].

Numerous natural ophiolites show evidence for supra-subduction zone magmatism in the upper plate following intra-oceanic subduction initiation, which results in thermally younger, thus thinner upper plates[8,45–47]. Our model does not account for such effects and hence may overestimate upper-plate thickness. Thinner upper plates would result in flatter continental subduction, which may lead to crustal decoupling and upper plate necking further away from the ophiolite front. If so, the size of the resulting far-travelled ophiolite sheets would be comparable to

the widest far-travelled ophiolite sheets in Oman and Anatolia (80–150 km; see Supplementary Note 1).

The current subduction of the Australian plate below the oceanic Banda arc provides an exciting example for a prospective future ophiolite belt. As more than 200 km of continental crust has already subducted below the oceanic plate[35], it most likely has reached eclogite facies conditions. Based on our model, decoupling of the Australian upper crust has already begun, or will begin in the geological near future. If decoupling is followed by nappe formation, the extrusion of the upper crust and simultaneous necking of the oceanic upper plate may lead to the emplacement of far-travelled ophiolites.

## Methods
**Numerical modelling.** The above-presented modeling results were obtained by solving the coupled set of non-linear thermo-mechanical equations. The steady-state momentum equation, the heat transfer equation and the incompressible mass conservation equations are formulated as:

$$\frac{\partial \tau_{ij}}{\partial x_j} - \frac{\partial P}{\partial x_i} = -\rho \mathbf{g}_i \tag{1}$$

$$\rho c_p \frac{DT}{Dt} = \frac{\partial}{\partial x_i}\left(k\frac{\partial T}{\partial x_i}\right) + Q_r + Q_s \tag{2}$$

$$\frac{\partial \mathbf{v}_i}{\partial x_i} = 0 \tag{3}$$

where $\mathbf{v}$ is the velocity vector, $T$ is the temperature, $k$ is the thermal conductivity, $\rho$ is the density, $c_p$ is the heat capacity at constant pressure, $Q_r$ is the radiogenic heat

production, $\tau$ is the deviatoric stress tensor, $\dot{\epsilon}$ is the deviatoric strain rate tensor, $P$ is the pressure, and $\mathbf{g}$ is the gravity acceleration vector. The visco-plastic dissipation or shear heating, $Q_s$, is expressed as:

$$Q_s = \tau_{ij}\left(\dot{\varepsilon}_{ij} - \dot{\varepsilon}_{ij}^e\right) \tag{4}$$

where $\dot{\varepsilon}_{ij}^e$ the elastic portion of the deviatoric strain rate.

The density field evolves according the following equation of state:

$$\rho = \rho_0(1 - \alpha(T - T_0))(1 + \beta(P - P_0)) \tag{5}$$

where $\rho_0$ is the reference density, $\alpha$ is the thermal expansivity, $\beta$ is the compressibility, $T_0$ and $P_0$ are the reference temperature and pressure which were respectively set to $0\,^\circ\mathrm{C}$ and $10^5$ Pa. We use the Boussinesq approximation; hence both density changes in the mass conservation equation and adiabatic heating are neglected.

The effective viscosity ($\eta$) relates to the deviator stress and strain rate tensor:

$$\tau_{ij} = 2\eta\dot{\varepsilon}_{ij} = \left(\frac{1}{\eta^v} + \frac{1}{\eta^e} + \frac{1}{\eta^p}\right)^{-1}\dot{\varepsilon}_{ij} \tag{6}$$

using a visco-elasto-plastic rheological model:

$$\dot{\varepsilon}_{ij} = \dot{\varepsilon}_{ij}^v + \dot{\varepsilon}_{ij}^e + \dot{\varepsilon}_{ij}^p \text{ where } \dot{\varepsilon}_{ij}^v = \dot{\varepsilon}_{ij}^{dis} + \dot{\varepsilon}_{ij}^{Peierls} \tag{7}$$

where the v, e, and p superscripts stand for viscous, elastic and plastic and the superscripts dis and Peierls correspond to dislocation and Peierls creep mechanisms.

The viscous strain rate account for contributions of both Peierls and dislocation creep. The contribution of dislocation creep is expressed as:

$$\dot{\varepsilon}_{ij}^{dis} = \dot{\varepsilon}_{II}^{dis}\frac{\tau_{ij}}{\tau_{II}} = \left(2A^{\frac{-1}{n}}f\,e^{\frac{Q}{nRT}}\right)^{-n}\tau_{II}^n\frac{\tau_{ij}}{\tau_{II}} \tag{8}$$

where $A$ is a pre-factor, $Q$ is the activation energy, $n$ is the stress exponent, $R$ is the universal gas constant, and $f$ is a correction factor[48]. The subscripts II stand for the square root of the second tensor invariant. The Peierls mechanism is taken into account in the mantle using the regularized formulation[49]. The Peierls strain rate is expressed as:

$$\dot{\varepsilon}_{ij}^{Peierls} = \dot{\varepsilon}_{II}^{Peierls}\frac{\tau_{ij}}{\tau_{II}} \tag{9}$$

where the second invariant strain rate is spelled as:

$$\begin{cases} \dot{\varepsilon}_{II}^{Peierls} = \left(2A^{Peierls}\right)^{-s}\tau_{II}^s\frac{\tau_{ij}}{\tau_{II}} \\ A^{Peierls} = f^{Peierls}\gamma\sigma^{Peierls}\left(E^{Peierls}e^{-\frac{(1-\gamma)^2Q^{Peierls}}{RT}}\right)^{\frac{-1}{s}} \\ s = \frac{Q^{Peierls}}{RT}(1-\gamma)^{(q-1)q\gamma} \end{cases} \tag{10}$$

where $s$ is an effective temperature-dependent stress exponent, $Q^{Peierls}$ is the activation energy ($=540$ kJ/mol), $\sigma^{Peierls}$ is the Peierls stress ($=8.5 \times 10^9$ Pa), $E^{Peierls}$ ($=5.7 \times 10^{11}$ s$^{-1}$), $q$ ($=2.0$), and $\gamma$ ($=0.1$)[50].

The elastic strain rate is written as:

$$\dot{\varepsilon}_{ij}^e = \frac{\tau_{ij}}{2G}\frac{\widehat{\tau}_{ij}}{\tau_{II}} \tag{11}$$

where $\tau_{ij}$ is the corotational time derivative of the stress tensor (Jaumann rate) and $G$ is the shear modulus ($=10^{10}$ Pa).

The plastic strain rate takes the form of:

$$\dot{\varepsilon}_{ij}^p = \dot{\varepsilon}_{II}^p\frac{\tau_{ij}}{\tau_{II}} \text{ with } \dot{\varepsilon}_{II}^p = \frac{F}{2\eta^{ve}} \text{ and is defined only if } F = \tau_{II} - C\cos\varphi - P\sin\varphi > 0 \tag{12}$$

where $\varphi$ is the friction angle and $C$ is the cohesion (see Table 1 for reference model values). The coefficient $\eta^{ve}$ is a visco-elastic coefficient dependent on both viscous and elastic moduli and on the time discretization[51]. No plastic strain softening was applied in the presented simulations.

The temperature is fixed at both the surface of the model ($0\,^\circ\mathrm{C}$) and the lower boundary ($1330\,^\circ\mathrm{C}$). No heat flows through the right and left boundaries. A plate convergence rate of 3 cm/year is achieved by prescribing constant normal inflow velocities of $|\mathbf{V_{in}}| = 1.5$ cm/year along the upper 140 km of the two model sides, while mass conservation is satisfied by gradually increasing outflow below 140 km. The shear stress is set to zero along the left, right, and lower boundaries. The upper boundary is a true free surface[18], thus its position evolves in response to tectonic loading. The initial topography is set to 0 km. The initial temperature field is computed by solving the heat transfer equation assuming steady state and neglecting visco-plastic dissipation using the reference thermal parameters (Table 1). During this initialization step, quasi-adiabatic mantle conditions are reached by assuming an artificially high conductivity within the asthenosphere.

The thermo-mechanical equations are solved using the finite difference/marker-in-cell technique[52]. The global linearized system of mechanical equations is solved with a direct-iterative scheme that combines both Powell-Hestenes iterations and Cholesky factorization[53]. Due to the non-linear nature of the considered rheological model, non-linear iterations are required at both local and global levels. At the local level, an exact partitioning of the strain rate tensor is obtained via successive Newton iterations[54,55]. At the global level, Picard iterations are used to best-satisfy mechanical equilibrium equations (to an absolute tolerance of $10^{-6}$ and within a maximum of 20 iterations).

**Model geometry**. The computational domain is a cross section of $1330 \times 410$ km. The model resolution is 1 km in both directions. The initial compositional geometry is inspired by reconstructions of pre-obduction geodynamic settings. It contains an oceanic domain (660 km wide) with a spreading ridge and a tilted weak zone in the center that ensures left-dipping intra-oceanic subduction initiation. The thermal structure of the oceanic lithosphere is calculated by applying a half-space cooling age model from 1.5 Myr at the center to 50 Myr at the edges of the ocean. The oceanic crust is 6 km thick and is overlain by a layer of uppermost sedimentary cover that linearly thickens from the ridge (0 km) towards the right edge of the continental domain (3 km). The transition from the continent to the ocean is defined by a passive margin geometry where the continental upper and lower crust linearly thin from 30 km to 5 km over the distance of 200 km. The uppermost sedimentary cover layer has a constant 3 km thickness over the continental domain.

## Data availability
Due to the very large size of the files, the model output data are available upon request from the corresponding author.

## Code availability
The numerical code used in this study is available from the main code developer (Thibault Duretz, e-mail: thibault.duretz@univ-rennes1.fr) upon reasonable request.

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

## Acknowledgements

The research leading to these results has received funding from the European Union's MSCA-ITN-ETN Project SUBITOP 674899. Numerical simulations were performed on the Utrecht University cluster Eejit. Thanks are due to Jeroen van Hunen and Cedric Thieulot for discussions that helped to initiate the project.

## Author contributions

K.P. conceived the research idea. T.D. and P.Y. designed the thermo-mechanical numerical code. A.A, P.Y., T.D., and K.P. designed the model setup. K.P. conducted the numerical simulations and interpreted the results together with P.Y., T.D., and E.W. All authors discussed the results and interpretations, and contributed to writing the paper.

## Competing interests

The authors declare no competing interests.
