## [Peer Review File · Nature Communications]

REVIEWER COMMENTS

Reviewer #1 (Remarks to the Author):

The manuscript made a very interesting read and investigates a very novel idea about extrusion of high-pressure rocks causing necking and thinning of the overlying ophiolite thrust sheet during subduction termination. The manuscript was very well and clearly written, easy to follow and very well illustrated which helps convey the idea to the reader. I agree with the idea of extruding high-pressure continental rocks which cause doming the over-riding ophiolite thrust sheet and I agree this maybe an important process in thinning the structurally higher-level rocks (in this case ophiolites) in many mountain belts/ accretionary systems. However, I unfortunately have some major points and problems with this model when applying it to explain 'far travelled' ophiolite thrust sheets which need to be thoroughly addressed and carefully considered in order to explain the field observations under such 'far travelled' ophiolite thrust sheets.

1. As the authors state, continental high-pressure metamorphic rocks associated with continental subduction associated with subduction termination occur under many ophiolite complexes around the world such as Oman, New Caledonia, Turkey, Cyclades ect. However, no-where in the manuscript does it mention the metamorphic sole rocks which directly underlie ophiolite thrust sheets. Metamorphic soles represent the subducted oceanic crust earlier in subduction history prior to arrival of the continental margin and occur everywhere under ophiolite thrust sheets when the complete sequence is exposed and were accreted to the upper plate as subduction continued. Metamorphic soles record upper amphibolite-granulite facies P-T conditions of 0.8-1.2 GPa (30-40 km depth) and 600-900 C, which are distinctly different from the continental HP-LT rocks described in this paper, and are often overlain by oceanic and continental rocks that have not experienced such high pressures (commonly non unmetamorphosed or low grade metamorphism). For example, in Oman, the metamorphic sole is overlain by the Haybi complex and Hawasna Units, which are overlain by unmetamorphosed shelf carbonates of the Arabian continental margin (e.g. Jabal al Adktar) which do not record high-pressure conditions. This suggests the Oman ophiolite was already emplaced onto the Arabian continental margin, in this case, and was associated with an entire history of thrust related deformation (e.g. Searle et al., 2004; Seale 2007), prior to high-pressure metamorphism of the leading edge of the Arabian continental margin (which occurs much structurally deeper e.g. As Sifah in Oman). Similar rock associations occur under most ophiolites suggesting they had to be tectonically emplaced prior to HP-LT metamorphism.

2. The cross sections in figure 1a are too overly simplified and not drawn correctly, (building on my previous point). The Saih Hatat dome in Oman is not all high-pressure rocks as drawn, it is mostly un-metamorphosed shelf carbonates, and it is only the deepest structural levels which reached high pressure conditions (carpholite to eclogite grade) see Fig. 14 in Searle, (2007). This poses a major problem to your model, which attempts to recreate extruding high-pressure rocks necking through the oceanic upper plate (ophiolite). The only way I can see this model sufficient to explain the real world geometry and account for the metamorphic sole and non-metamorphic nappes exposed beneath the ophiolite is if you had the upper plate comprising of the ophiolite, sole and non-metamorphic nappes and the ophiolite was at least partially emplaced onto the continental margin (associated with the metamorphic sole and un-metamorphosed rocks beneath the ophiolite). If all these nappes were accreted into an upper plate position and then moved the location of decoupling to deeper levels within the continental margin (i.e. the zone of active thrusting/subduction stepping down section). This would cause burial of the structurally deeper continental margin to high-pressures beneath the ophiolite and sole and unmetamorphosed continental margin (now in the upper plate) before the HP-LT rocks extruded and neck the overlying upper plate (ophiolite plus sole plus shelf carbonates ect.). However, the problem here is that you still require the ophiolite to be a far travelled nappe as it must have been emplaced at least beyond the locus of subduction.

3. In Oman extrusion of subducted crust does not explain far travelled ophiolites because the HP – UHP rocks are 15 m.y. later than the formation of the ophiolite and the amphibolites of the Metamorphic sole rocks. The HP rocks are not coeval with far-travelling thrust emplacement. In Oman the ophiolite was formed at 96-95 Ma and the sole was simultaneous (95-94 Ma). The

ophiolite was obducted more than 150 km across the continental margin 15 m.y. before the attempted subduction of the margin and HP metamorphism (79 Ma; Warren et al., 2003, 2005). The structure and timing constraints in Oman are well understood (Searle et al., 1994; Warren et al., 2003, 2005; Agard et al. 2010; Searle 2007; Rioux et al. 2016). There is no explanation of the granulite facies thrust sheets beneath the ophiolite in the north (Bani Hamid) either.

4. In your model, you initiate the subduction zone at an oceanic detachment fault. What is the reasoning for this? I realise you discuss that this may not be realistic but how might differences in oceanic lithosphere age change the model result? In many metamorphic soles, we often find the age of the subducted oceanic crust to be older and geochemically different suggesting the subducted crust formed at a different time and setting to the ophiolite in the upper plate.

5. In your model, you discuss how extrusion of lower plate HP material causes necking and thinning of the ophiolite but you don't discuss the paradox of metamorphic soles which often record pressures equating to depths of 30-40 km, over double the current thickness of the over-riding ophiolites. I'm just wondering if the extrusion of high-pressure lower plate material and necking of the upper plate idea might be able to explain this observation? Based on my arguments above, it would require the ophiolite to already be at least partly emplaced onto the continental margin (i.e. above the zone of decoupling).

6. Why do you choose only one convergence velocity? How does convergence velocity change the end result? I think this needs to be better documented and explored in parameter space including subduction dip angles. However I feel there are more fundamental problems with the idea the model is based upon (see above comments).

7. Why in the model does the upper continental crust decrease in subduction velocity while the lower crust continues to subduct? This implies the upper and lower crust are mechanically decoupled (presumably from your initial starting conditions and rheology). This needs further explanation. I agree the upper and lower crust have different rheology's but what parameters do you choose and why?

Further comments:

Line 22: Ophiolites are rooted in the suture between 2 continental plates in many mountain belts, not necessarily separated from the ocean. This needs to be clarified for the ill informed reader.

Line 32: Yes this is strictly correct, but they also have non metamorphosed oceanic and continental rocks underneath them, which explain the earlier stages of obduction history prior to HP-LT metamorphism (way structurally deeper).

Line 177-181: Why exactly does shear heating cause such big changes in model output, what are the differences in temperature and the effect on rheology? This is not clear to the reader.

Line 212: This is not absolutely correct. Guillmette et al., (2018) garnet ages in the Oman metamorphic sole now suggest subduction commenced 8 Myr before the ophiolite formed, also in Western Turkey, e.g. Plunder et al., (2015) show the HP-LT metamorphism occurred as the sole itself was exhuming i.e. cooling happened immediately.

Line 236: See my previous comment about metamorphic soles and the preservation of non-metamorphosed continental margin rocks beneath ophiolites, this must be carefully taken into account as these geological relationships suggest the ophiolites are far travelled nappes.

Line 242-244: Yes this explains what we see in many HP terranes which undergo diverse P-T-t paths, but this is nothing new.

Line 250: Ophiolite metamorphic soles need to be taken into account here. They suggest the thickness of the upper plate at the time of subduction was 30-40 km thick, although present day ophiolite thicknesses are less than half this.

Line 253: Missing a reference here?

Reviewer #2 (Remarks to the Author):

General comments

The paper applies numerical geodynamic modelling to a system involving subduction of a continental margin beneath an oceanic plate, with the aim of testing ideas about nappe formation and the distribution of large ophiolites separated from their root zones by some unspecified distance. The results outline a series of conditions under which the upper part of the subducting continental plate can detach from the lower part and extrude from the subduction zone in the form of nappes, on which parts of the upper oceanic plate can ride for some unspecified distance.

I am not a modeller and cannot comment on the suitability of the assumptions that go into the model. However, I am a "consumer" of such models and my comments below attempt to show how the justification and presentation of the model and the results themselves might be improved to be useful to people like me.

Part of the problems stem from the attempt to present the work in the framework of a short paper, with limitations on space. The figures, which are the essential part of the paper, are elegant but they are simply too small to very useful to the probable consumers. The presentation of the problem, and the explanation of what is actually happening in the figures, seems to be oriented very much toward other modellers, rather than geologists who actually work on the problems involved. The authors and the editors need to consider whether that is really useful, and think about making this more accessible to people like me (and I have worked previously on crustal subduction, eclogites, nappes, etc).

Despite the overall excellence of the English (if not the style), there are quite a few grammatical errors and typos that need fixing.

Specific comments

l. 38. This is the first mention of "far-travelled" ophiolites, although this is, judging from the title, the main focus of the paper. So you need to introduce this concept right at the start, along with a definition of what you mean by "far".

l. 58-70. Unfortunately this introduction to the problem is hard to follow even for someone who has worked in nappes. It needs a clearer explanation of what is involved in the nappe formation and the emplacement of the ophiolite, to set up the problem for modelling. With regard to lines 58-60, is the distance travelled (measured from where?) not part of the list of things to be accounted for?

Fig. 2. It appears, especially in the panels c-e, that the model has similar lithospheric thickness beneath the continental plate and the oceanic one. This is an unlikely scenario, and do you think it affects the dynamics of your model?

l. 102. What is meant by "consummation"?

l. 113. Why would the burial velocity slow down when the leading edge of the plate reaches eclogite facies? Wouldn't you think that it would accelerate because that part of the plate increases in density?

l. 117. Why is upper-plate extension "gravity-driven" extension? Where are the gravitational forces being applied? Is not the uplift itself driving extension?

Figures 2 and 3. Most of the action in these models is going on in the upper part of the figure, and these are simply too small to let the reader really see what is going on. While these look elegant,

the information in them is difficult to access. I am not sure what sort of redesign will solve this problem, but without it the only readers probably will be other modellers. One possibility is to use the page space to show the relevant parts of Figures 2 and 3 (the left panels, cropped) and put the right-hand panels in the supplementary data.

I. 136. Again, will the beginning of eclogitization not decrease the buoyancy force? Is this part of the model parameters, and if not, why not?

I. 147. Why does nappe formation result in uplift? Isn't it the other way around? The features mentioned in this paragraph are not obvious in Fig. 3a (at least to me). Again, this reflects the nature of the figure, on which it is difficult to point out these features because of the small size of the figure.

I. 154. If you have explained why this acceleration occurs, I have missed it.

WL Griffin

Reviewer #3 (Remarks to the Author):

The work of Porkolab et al. addresses in details the mode of exhumation of continental lithosphere responsible for the fragmentation of ophiolitic sheets through advanced numerical models at the lithospheric scale. The results of the models are impressive, and provide a very detailed framework for the interpretation of the tectonic processes observed on the field. The presentation of the results is very clear and will be useful for a wide audience, from undergraduate students to senior researchers. The impact of this study could even be wider than people working on ophiolites, as the processes of nappe exhumation modeled in this paper also occur in the most emblematic mountain belts such as the Himalaya or the European Alps. This paper indeed demonstrates (in the 'key parameters...' section) how nappe exhumation may occur without involving tectonic-climate interactions (as required in analogue experiments to reproduce the same pattern of nappe stacking and exhumation).

I therefore recommend publication of this paper as it is, as I think it is the most valuable work I have read so far in Nature communications. Congratulations to the authors for this very inspiring work.

Reviewer comments and replies for manuscript NCOMMS-20-28739

Below we provide our response to the comments of the reviewers (in bold) and detail the modifications made to improve our manuscript. When referring to a figure, we generally mean the figure in the new revised version of the manuscript. We also use an additional figure in this document as part of our reply. When referring to that figure, we always use 'Fig. 1 in this document'.

Reviewer 1

The manuscript made a very interesting read and investigates a very novel idea about extrusion of high-pressure rocks causing necking and thinning of the overlying ophiolite thrust sheet during subduction termination. The manuscript was very well and clearly written, easy to follow and very well illustrated which helps convey the idea to the reader. I agree with the idea of extruding high-pressure continental rocks which cause doming the over-riding ophiolite thrust sheet and I agree this maybe an important process in thinning the structurally higher-level rocks (in this case ophiolites) in many mountain belts/ accretionary systems. However, I unfortunately have some major points and problems with this model when applying it to explain 'far travelled' ophiolite thrust sheets which need to be thoroughly addressed and carefully considered in order to explain the field observations under such 'far travelled' ophiolite thrust sheets.

Comment 1. As the authors state, continental high-pressure metamorphic rocks associated with continental subduction associated with subduction termination occur under many ophiolite complexes around the world such as Oman, New Caledonia, Turkey, Cyclades ect. However, nowhere in the manuscript does it mention the metamorphic sole rocks which directly underlie ophiolite thrust sheets. Metamorphic soles represent the subducted oceanic crust earlier in subduction history prior to arrival of the continental margin and occur everywhere under ophiolite thrust sheets when the complete sequence is exposed and were accreted to the upper plate as subduction continued. Metamorphic soles record upper amphibolite-granulite facies P-T conditions of 0.8-1.2 GPa (30-40 km depth) and 600-900 C, which are distinctly different from the continental HP-LT rocks described in this paper, and are often underlaid by oceanic and continental rocks that have not experienced such high pressures (commonly non unmetamorphosed or low grade metamorphism). For example, in Oman, the metamorphic sole is underlaid by the Haybi complex and Hawasna Units, which are underlaid by unmetamorphosed shelf carbonates of the Arabian continental margin (e.g. Jabal al Adktar) which do not record high-pressure conditions. This suggests the Oman ophiolite was already emplaced onto the Arabian continental margin, in this case, and was associated with an entire history of thrust related deformation (e.g. Searle et al., 2004; Seale 2007), prior to high-pressure metamorphism of the leading edge of the Arabian continental margin (which occurs much structurally deeper e.g. As Sifah in Oman). Similar rock associations occur under most ophiolites suggesting they had to be tectonically emplaced prior to HP-LT metamorphism.

We thank the reviewer for the comments, which reveal that our manuscript needs improvements on clarity of the used terminology, the level of details in the introduction, and the model to nature comparisons. Careful evaluation of the comments ascertains that we have no actual disagreement with the reviewer on the evolution of obduction systems. However, we use a different terminology and have a different point of view, which caused some misunderstanding. For this reason, we improved on text and figures to ensure that the terminology we use is well-defined and the comparison with natural ophiolite sites is spelled out

clearer. Below in this document we detail the typical evolution of obduction systems (Fig 1 in this document) using the Oman example, and show that our model is relevant for the Oman ophiolite belt as well as many others. We provide our answer to this major comment in three sub-points (a-c), addressing different parts of the comment.

[Redacted]

Fig. 1 Evolutionary steps of obduction systems based on the Oman case (after Searle and Cox, 1999). The evolutionary steps are compared to equivalent snapshots of our reference model. **a** Intra-oceanic subduction with the formation of the metamorphic sole. **b** Continental subduction below the oceanic upper plate (i.e. obduction) and the accretion of non-metamorphic sedimentary cover units to the upper plate. **c** Extrusion of the subducted (HP-LT) continental crust. The conditions of this crustal extrusion and its role in the deformation of the oceanic upper plate is the focus of our paper. **d** Present-day cross section through Oman (after Searle, 2007) (Fig. 1a in the revised manuscript). **e-h** Equivalent snapshots of our reference model. The last snapshot shows the near-surface structure and composition of the reference

model at the final time step. The color code of this section is the same as for the Oman cross section (d) to facilitate the comparison.

a) Metamorphic soles

Metamorphic soles are formed during the early phase of intra-oceanic subduction, prior to the subduction of the continental margin. We completely agree with the reviewer's description and interpretation of metamorphic soles and are aware of all the mentioned literature and information. However, we wish to stress that the formation and characteristics of metamorphic soles are not the key points of our study. In this paper, we focus on the subduction and extrusion of the *continental crust*, which is indeed a distinctly different (later) stage in the evolution of obduction systems, as also stated by the reviewer (Fig 1 in this document). In our manuscript, we show the initial model conditions (Fig 2a), and after the initial stage of continental subduction, i.e. obduction (Fig 2b). In between these steps, intra-oceanic subduction occurs, which is the stage where metamorphic soles are formed in nature (Figs 1a, e in this document). We do not show a snapshot of this evolutionary stage in our paper, because we do not aim to model metamorphic sole formation which is yet another intriguing topic. We focus entirely on continental subduction-exhumation dynamics underneath an oceanic upper plate (obduction *sensu stricto*). We believe that the main message of the paper would receive less emphasis if we discussed all the evolutionary steps of obduction systems with the same detail. On the other hand, we see the importance of being more precise and complete in our description of ophiolite belts, hence we decided to include additional information in the introduction of the manuscript including the formation of metamorphic soles.

b) Obduction vs intra-oceanic subduction and far-travelled ophiolite sheets

Another issue that probably caused misunderstanding is the different use of terminology, and the lack of clear definitions in the introduction of our manuscript. We are under the impression that the reviewer defines intra-oceanic subduction already as obduction, while we reserve the term 'obduction' for when the continental crust subducts below an oceanic upper plate. As such for the reviewer, the emplacement of the ophiolite already occurs when intra-oceanic subduction begins, while for us it only occurs when the continental crust starts to subduct (Fig 2b, Figs 1b and f in this document). Due to this difference in terminology, we have a different view on when an ophiolite sheet is *emplaced*. In our view, the emplacement of the ophiolite occurs when the continent starts to subduct below the oceanic upper plate, while the future ophiolite sheet is formed earlier in an intra-oceanic setting. One may argue (and if we are correct, this is the argument of the reviewer), that the emplacement of ophiolite sheets on top of continents is thus simply the result of continental subduction, and extrusion of the crust is not needed to explain emplacement. We agree, that *initial emplacement* of the ophiolite on top of the continent does not require extrusion of the crust. However, this process does not explain the breaking of the oceanic upper plate and the emplacement of the *far-travelled ophiolite sheet*; the piece of oceanic lithosphere that is detached from the rest of the oceanic domain. We show in this paper, that a piece of the oceanic upper plate is broken off and transported further on top of the continent via roof thrusting, due to the extrusion of the subducted continental crust (See Fig. 3 and 6). This is the process that explains the structure of ophiolite belts (an ophiolite sheet detached from the rest of the oceanic domain, what we call *far-travelled ophiolite sheet*). In this sense, the emplacement of the oceanic lithosphere on top of the continent is simply explained by continental subduction, but the emplacement of the *far-travelled ophiolite sheet* is only explained by crustal extrusion, following subduction. This is the main point we wish to make. We noted the need for more clarity regarding the terminology, and so we re-formulated our abstract and introduction to make all the expressions that we use well-defined from the beginning.

c) Non-metamorphic passive margin formations below the ophiolite

The reviewer raises the point that non-metamorphic sedimentary units are found below the Oman ophiolite, and above the subducted and exhumed HP-LT continental units. We made the same point in our manuscript, and highlighted the accretion of sedimentary cover units to the oceanic upper plate during oceanic, and later continental subduction in our model (Lines 111-113 and 207-208). The majority of the sedimentary cover does not experience subduction in our model, but is accreted to the upper plate when extrusion of the HP-LT crust occurs. Thus, the presence of non-metamorphosed sedimentary cover units below the Oman ophiolites and many other ophiolite sites can clearly be explained by our model (see also Fig 1 in this document). Furthermore, the reviewer emphasizes the presence of further non-metamorphic passive margin units below the stacked sedimentary cover units (i.e. below the Haybi complex and Hawasina Units in case of Oman). In our reference model, the stacked and accreted uppermost sedimentary cover units (pink unit) are structurally underlain by non-metamorphic to low-grade continental crust (Fig 1h in this document, Fig 6), that did not subduct deep enough below the upper plate to experience HP-LT metamorphism, and thus can be considered as analogous to the mentioned passive margin formations (Paleozoic to Mesozoic basement and sediments that are not incorporated in the Haybi and Hawasina units). To ensure that all the near-surface geological structures and units produced by our model are clearly defined, we added an extra figure focusing on the final geometry of our model (Fig 6). This figure contains all the necessary information for the reader to correlate the geological units in our model with geological units in natural ophiolite belts.

In summary, we did our best to add information, figures, and clarity in our revised manuscript following the points raised by the reviewer. We hope that all of them are now satisfied.

Comment 2. The cross sections in figure 1a are too overly simplified and not drawn correctly, (building on my previous point). The Saih Hatat dome in Oman is not all high-pressure rocks as drawn, it is mostly un-metamorphosed shelf carbonates, and it is only the deepest structural levels which reached high pressure conditions (carpholite to eclogite grade) see Fig. 14 in Searle, (2007). This poses a major problem to your model, which attempts to recreate extruding high-pressure rocks necking through the oceanic upper plate (ophiolite). The only way I can see this model sufficient to explain the real world geometry and account for the metamorphic sole and non-metamorphic nappes exposed beneath the ophiolite is if you had the upper plate comprising of the ophiolite, sole and non-metamorphic nappes and the ophiolite was at least partially emplaced onto the continental margin (associated with the metamorphic sole and un-metamorphosed rocks beneath the ophiolite). If all these nappes were accreted into an upper plate position and then moved the location of decoupling to deeper levels within the continental margin (i.e. the zone of active thrusting/subduction stepping down section). This would cause burial of the structurally deeper continental margin to high-pressures beneath the ophiolite and sole and unmetamorphosed continental margin (now in the upper plate) before the HP-LT rocks extruded and neck the overlying upper plate (ophiolite plus sole plus shelf carbonates ect.). However, the problem here is that you still require the ophiolite to be a far travelled nappe as it must have been emplaced at least beyond the locus of subduction.

We thank the reviewer to have pointed out this issue. Indeed, we made a mistake in the first version of the manuscript when drawing the cross section of Oman. The southwestern side of the Saih-Hatat dome consists of non-metamorphic, Mesozoic passive margin succession, that is structurally below the ophiolite and the accreted sediments. The mistake is now corrected. Moreover, the sections of Oman and New

Caledonia were also re-designed to show more geological details (Figs 1a and b). However, the presence of the non-metamorphic units below the upper plate formations do not pose any problem to our model. As already explained above (Comment 1/c), low-grade to non-metamorphic rocks in the dome are captured by our model (see Fig 1 in this document and/or Fig 6 in the revised manuscript). As shown by Fig 1h in this document, the window of exhumed crust shows a clear transition from blueschist-eclogite facies rocks to non-metamorphic rocks towards the far-travelled ophiolite sheet, just like in Oman as well as other ophiolite belts in nature. The notion of the reviewer that *“The only way I can see this model sufficient to explain the real world geometry and account for the metamorphic sole and non-metamorphic nappes exposed beneath the ophiolite is if you had the upper plate comprising of the ophiolite, sole and non-metamorphic nappes”* describes the actual situation in our model very well. The upper plate indeed consists of the ophiolite, and the accreted non-metamorphic sediments, already emplaced on the continental margin in our model (see Figs 2, 3, and Fig 1 in this document). The extruding continental upper crust cause necking of this composite upper plate in our model. In our opinion we have no opposing views regarding this matter.

Comment 3. In Oman extrusion of subducted crust does not explain far travelled ophiolites because the HP – UHP rocks are 15 m.y. later than the formation of the ophiolite and the amphibolites of the Metamorphic sole rocks. The HP rocks are not coeval with far-travelling thrust emplacement. In Oman the ophiolite was formed at 96-95 Ma and the sole was simultaneous (95-94 Ma). The ophiolite was obducted more than 150 km across the continental margin 15 m.y. before the attempted subduction of the margin and HP metamorphism (79 Ma; Warren et al., 2003, 2005). The structure and timing constraints in Oman are well understood (Searle et al., 1994; Warren et al., 2003, 2005; Agard et al. 2010; Searle 2007; Rioux et al. 2016). There is no explanation of the granulite facies thrust sheets beneath the ophiolite in the north (Bani Hamid) either.

This comment is also rooted in the same terminological difference that we explained with reference to Comment 1; we define the initial emplacement of the ophiolites as the moment when the continental crust starts subducting below the oceanic upper plate, while the reviewer regards the intra-oceanic subduction stage already as obduction. We are certain that we have no actual disagreement, as detailed below:

The cited literature and the related evolutionary model of Oman fits very well our interpretation. 1) intra-oceanic subduction at ca. 95 Ma with the formation of the metamorphic sole (which is not in our focus); 2) oceanic subduction and the accretion of the sedimentary cover units; 3) initiation of continental subduction (=obduction) at ca. 85 Ma and HP-LT metamorphism from ca. 80 Ma; 4) exhumation (extrusion) of the subducted crust and final emplacement of the far-travelled ophiolite sheet at ca. 70 Ma. From the initiation of continental subduction (ca. 85 Ma) to the exhumation (ca. 70 Ma), the time span is 15 Myr. This is our best estimate for the continental subduction-exhumation cycle in Oman, and this is the 15 Myr plotted on Fig 1d in the original manuscript. In our reference model, the same evolution from continental subduction initiation to the exhumation of the HP units takes ca. 17 Myr, which is closely comparable.

4. In your model, you initiate the subduction zone at an oceanic detachment fault. What is the reasoning for this? I realise you discuss that this may not be realistic but how might differences in oceanic lithosphere age change the model result? In many metamorphic soles, we often find the age of the subducted oceanic crust to be older and geochemically different suggesting the subducted crust formed at a different time and setting to the ophiolite in the upper plate.

We were aiming for a simple setup, without focusing too much on the mechanism of subduction initiation, which is a broad topic on its own. Here we focus on the process of continental subduction-exhumation and therefore did not wish to enter any debate on intra-oceanic subduction initiation processes. The reason for the chosen configuration is that by applying the inclined weak zone, a favorably dipping oceanic subduction is achieved very smoothly, and allows us to focus on our question of interest. At the same time, the significance of oceanic detachments in intra-oceanic subduction initiation has also been demonstrated (e.g. Maffione et al., 2015).

5. In your model, you discuss how extrusion of lower plate HP material causes necking and thinning of the ophiolite but you don't discuss the paradox of metamorphic soles which often record pressures equating to depths of 30-40 km, over double the current thickness of the over-riding ophiolites. I'm just wondering if the extrusion of high-pressure lower plate material and necking of the upper plate idea might be able to explain this observation? Based on my arguments above, it would require the ophiolite to already be at least partly emplaced onto the continental margin (i.e. above the zone of decoupling).

This is an interesting point and might be worthwhile to pursue with a different model setup, but not in this contribution since here we focus on obduction (*sensu stricto*). Our goal here is merely to show the dynamics of continental subduction and extrusion, and how that leads to the breaking of the upper plate and far-travelled ophiolite sheet emplacement. This is the single message that we aim to deliver in this paper, and we would like to avoid discussing the characteristics and paradoxes of metamorphic soles, not to divert attention from our main message. Additionally, for another study, an alternative pressure-depth conversion to the lithostatic assumption might be worthwhile to consider to explain such enigmatic data. The dynamic pressure component recorded by metamorphic rocks might lead to substantial overestimation of the depth in a variety of geodynamic environments when using lithostatic pressure-depth conversion (Moullas et al., 2014).

6. Why do you choose only one convergence velocity? How does convergence velocity change the end result? I think this needs to be better documented and explored in parameter space including subduction dip angles. However, I feel there are more fundamental problems with the idea the model is based upon (see above comments).

Convergence velocity is indeed a very important parameter in geodynamic models that affects the forcing on tectonic plates and influences key parameters for deformation processes, such as strain rate. We chose a conservative 3 cm/yr convergence velocity aiming to be close to average values of relevant convergent systems that we have studied. We have taken the point raised by the reviewer, and decided to include a velocity test in the Supplementary Information file (see Supplementary Fig. 3). The velocity test shows that minor changes (± 1 cm/yr) in convergence velocity do not have a dramatic effect on model results, but larger changes ($+2$ cm/yr) do affect the outcome significantly, explained by changes in strain rate. This is further detailed in the new version of the manuscript (in Supplementary Note 2).

7. Why in the model does the upper continental crust decrease in subduction velocity while the lower crust continues to subduct? This implies the upper and lower crust are mechanically decoupled (presumably from your initial starting conditions and rheology). This needs further explanation. I agree the upper and lower crust have different rheology's but what parameters do you choose and why?

Indeed, the upper and lower continental crust are decoupled in our reference model and are of key importance in the process of extrusion. The reviewer's questions regarding the used parameters and the role of coupling were already extensively included in the original manuscript. In the 'Modeling strategy' section we wrote: '*The continental basement is divided in two parts (upper and lower crust) constituted by different materials (Fig. 2a and Table 1), which introduces a decoupling level at the base of the upper crust (Fig. 2a).*' We provided every relevant parameter for the upper and lower crust rheologies (Table 1), strength profiles (Fig. 2a), and further rheology tests (Fig. 4) to show the importance of coupling/decoupling. We show that depending on the choice of crustal flow laws and thermal parameters, the coupling and hence the extrusion of the crust can be substantially different (Fig 4 and Supplementary Fig. 2). In our view any additional explanation on the rheological setup of the continental crust would be a repetition in the manuscript.

Further comments:

Line 22: Ophiolites are rooted in the suture between 2 continental plates in many mountain belts, not necessarily separated from the ocean. This needs to be clarified for the ill informed reader.

The reviewer is right, it is either an ocean or the suture zone of a former ocean, as also emphasized in the introduction (line 43 in the revised manuscript). Based on the comment, we changed the expression from 'oceanic domain' to 'root' which is suitable for referring to both cases (suture zone or open ocean).

Line 32: Yes this is strictly correct, but they also have non metamorphosed oceanic and continental rocks underneath them, which explain the earlier stages of obduction history prior to HP-LT metamorphism (way structurally deeper).

We agree and are aware of the presence of non-metamorphic units below the ophiolites (as clarified above at Comments 1-2), but here we would like to emphasize the presence of HP units.

Line 177-181: Why exactly does shear heating cause such big changes in model output, what are the differences in temperature and the effect on rheology? This is not clear to the reader.

Point taken. We added a new figure to the manuscript (Fig. 5) that aims to provide additional information on the effect of shear heating. We plotted the energy dissipation resulting from shear heating prior to and after the localization of the main thrust (Figs. 5 a, b, left panels). Results show that shear zone localization results in heat dissipation that is 2 orders of magnitude higher than radiogenic heat production in the continental crust. On the right panels, we show the viscosity for the same snapshots to highlight how the crustal material was weakened in response to shear heating.

Line 212: This is not absolutely correct. Guillmette et al., (2018) garnet ages in the Oman metamorphic sole now suggest subduction commenced 8 Myr before the ophiolite formed, also in Western Turkey, e.g. Plunder et al., (2015) show the HP-LT metamorphism occurred as the sole itself was exhuming i.e. cooling happened immediately.

This is the same misunderstanding as explained above (Comments 1-3): we are focusing on the *continental* subduction-exhumation cycle, not counting the formation of the metamorphic sole during

intra-oceanic subduction. The term ‘continental subduction-exhumation cycle’ was added here to avoid confusion.

Line 236: See my previous comment about metamorphic soles and the preservation of non-metamorphosed continental margin rocks beneath ophiolites, this must be carefully taken into account as these geological relationships suggest the ophiolites are far travelled nappes.

See answers to Comments 1-3.

Line 242-244: Yes this explains what we see in many HP terranes which undergo diverse P-T-t paths, but this is nothing new.

Indeed, the importance of rheology in the coupling/decoupling of a subducting lithosphere has already been demonstrated, thus our results merely support previous findings and extend their applications to obduction systems. That is why we did not claim it was new, we only wrote “*Our results also support ...*”.

Line 250: Ophiolite metamorphic soles need to be taken into account here. They suggest the thickness of the upper plate at the time of subduction was 30-40 km thick, although present day ophiolite thicknesses are less than half this.

See reply to Comment 5.

Line 253: Missing a reference here?

Pont taken, we added a reference to Supplementary Note 1, where the ophiolite sheet width measurements are detailed and referenced.

Reviewer 2

General comments

The paper applies numerical geodynamic modelling to a system involving subduction of a continental margin beneath an oceanic plate, with the aim of testing ideas about nappe formation and the distribution of large ophiolites separated from their root zones by some unspecified distance. The results outline a series of conditions under which the upper part of the subducting continental plate can detach from the lower part and extrude from the subduction zone in the form of nappes, on which parts of the upper oceanic plate can ride for some unspecified distance.

I am not a modeller and cannot comment on the suitability of the assumptions that go into the model. However, I am a “consumer” of such models and my comments below attempt to show how the justification and presentation of the model and the results themselves might be improved to be useful to people like me.

Part of the problems stem from the attempt to present the work in the framework of a short paper,

with limitations on space. The figures, which are the essential part of the paper, are elegant but they are simply too small to very useful to the probable consumers. The presentation of the problem, and the explanation of what is actually happening in the figures, seems to be oriented very much toward other modellers, rather than geologists who actually work on the problems involved. The authors and the editors need to consider whether that is really useful, and think about making this more accessible to people like me (and I have worked previously on crustal subduction, eclogites, nappes, etc).

Despite the overall excellence of the English (if not the style), there are quite a few grammatical errors and typos that need fixing.

We thank the reviewer for the comments. We considered the remarks about making our illustrations larger and our descriptions more extensive to provide better visibility and better potential applicability for the field geology community. We took several actions in order to achieve that. 1) We enlarged the panels of Fig. 3 and re-arranged it in a full-page format. This way the near-surface objects (structural interpretation, velocity vectors, strain rate field, stress orientations, and second stress invariant field) are much more visible; 2) We added a figure that focuses on the effect of shear heating to better explain the process of strain localization and nappe formation which is certainly an interesting aspect for the geology community (Fig. 5); 3) We added a figure that focuses on the near-surface geometry, structure, and composition of the final snapshot or our reference model (Fig. 6). This figure will certainly allow for a much better comparison with ophiolite sites in nature and help field geologists to make use of our work; 4) We extended and clarified the introduction, the description of model results, and the ‘Comparison to natural ophiolite belts’ section to provide all the necessary information for model consumers. We are convinced that the geology community will find great use of our work when seeking physics-based explanations for field observations.

Specific comments

Comment 1. l. 38. This is the first mention of “far-travelled” ophiolites, although this is, juggling from the title, the main focus of the paper. So you need to introduce this concept right at the start, along with a definition of what you mean by “far”.

We agree with this point. We realized (also based on the misunderstandings with reviewer 1) that we need to improve on the clarity of the term used in our manuscript. In the revised version of the manuscript, we now define all important terminology that we use at the beginning to avoid any further confusion. In this case, we already define ‘far-travelled’ in the second sentence of the abstract: *‘Numerous ophiolite belts on Earth exhibit a far-travelled ophiolite sheet that is separated from its oceanic root by tectonic windows of continental crust that experienced subduction-related high pressure-low temperature (HP-LT) metamorphism during obduction.’*

Comment 2. l. 58-70. Unfortunately this introduction to the problem is hard to follow even for someone who has worked in nappes. It needs a clearer explanation of what is involved in the nappe formation and the emplacement of the ophiolite, to set up the problem for modelling. With regard to lines 58-60, is the distance travelled (measured from where?) not part of the list of things to be accounted for?

The section in question was rephrased in order to provide a better introduction to the modelling problem (lines 69-77 in the revised manuscript). Regarding the definition of far-travelled ophiolite sheets: we

define it as a piece of oceanic lithosphere that was detached from its oceanic root and further transported on top of the continent. This can be a distance of tens or hundreds of kilometers, depending on the amount of roof thrusting. For this reason, we prefer not to specify a distance that is required to call a sheet ‘far-travelled’, it simply needs to be detached from the root, separated from it by the extruding continental units. We rephrased our introduction (and abstract) to make this definition as clear as possible.

Comment 3. Fig. 2. It appears, especially in the panels c-e, that the model has similar lithospheric thickness beneath the continental plate and the oceanic one. This is an unlikely scenario, and do you think it affects the dynamics of your model?

The continental and oceanic plates have different thicknesses in our models. The scale on Fig. 2 (black ticks every 100 km) shows that the ocean (even at the older parts) is less than 100 km thick, while the continent is a bit less than 150 km (precisely 140 km) thick.

Comment 4. l. 102. What is meant by “consummation”?

By consummation we meant subduction. We changed it to subduction in the revised manuscript.

Comment 5. l. 113. Why would the burial velocity slow down when the leading edge of the plate reaches eclogite facies? Wouldn't you think that it would accelerate because that part of the plate increases in density?

We thank the reviewer for raising this interesting point. Metamorphic phase transitions are not included in the model calculations; therefore, we cannot take eclogitization into account. However, even if metamorphic phase transitions were included in the model calculations, the high-pressure felsic continental crust would still be lighter than the surrounding mantle. Hence, subduction would still be slowing down, only perhaps less quickly than in our model.

Comment 6. l. 117. Why is upper-plate extension “gravity-driven” extension? Where are the gravitational forces being applied? Is not the uplift itself driving extension?

Nappe formation in the subducted crust induces uplift of the upper plate. Uplift increases the gravitational potential (gravity calculations are included in the model, so when the topography rises, gravitational potential will increase accordingly), which results in local extension (vertical σ_1) despite the overall convergence. The term ‘gravity-driven’ is used to emphasize that no kinematic boundary conditions was changed to induce extension, but it is the result of the internal forcing of the obduction system during convergence. One could also call it ‘uplift-driven’, however, gravity-driven is the commonly used expression for this phenomena in the literature.

Comment 7. Figures 2 and 3. Most of the action in these models is going on in the upper part of the figure, and these are simply too small to let the reader really see what is going on. While these look elegant, the information in them is difficult to access. I am not sure what sort of redesign will solve this problem, but without it the only readers probably will be other modellers. One possibility is to use the page space to show the relevant parts of Figures 2 and 3 (the left panels, cropped) and put the right-hand panels in the supplementary data.

We enlarged the panels of Fig. 3 and re-arranged it in a full-page format. This way the near-surface objects (structural interpretation, velocity vectors, strain rate field, stress orientations, and second stress invariant field) are much more visible. We also managed to slightly enlarge the right side panels of Fig. 2,

however, a complete re-design was not possible as the left panels (P-T evolution) are key for the understanding of the main messages, and therefore cannot be moved to the supplement. Additionally, the new Fig. 6 will allow a very clear final view of the model geometry.

Comment 8. l. 136. Again, will the beginning of eclogitization not decrease the buoyancy force? Is this part of the model parameters, and if not, why not?

See answer to Comment 6. Eclogitisation process (in terms of phase changes) is not included in our model.

Comment 9. l. 147. Why does nappe formation result in uplift? Isn't it the other way around? The features mentioned in this paragraph are not obvious in Fig. 3a (at least to me). Again, this reflects the nature of the figure, on which it is difficult to point out these features because of the small size of the figure.

We do not quite understand the first part of this comment. Thrusting and nappe formation lead to the thickening of the low-density continental crust, therefore result in surface uplift (mountain building). For the second part, we solved this by enlarging Fig. 3, as detailed above.

Comment 10. l. 154. If you have explained why this acceleration occurs, I have missed it.

Point taken; we added explanation on the acceleration of extrusion by rephrasing the sentence. Extrusion is initially slow, because the shear zones are not established yet, and the upper plate is still coherent above the subducted crust. Once the shear zones are localized, and the upper plate is dissected by the extensional shear zones, most of the strain will be localized at the structures accommodating the extrusion. This results in the acceleration of extrusion.

Reviewer 3

The work of Porkolab et al. addresses in details the mode of exhumation of continental lithosphere responsible for the fragmentation of ophiolitic sheets through advanced numerical models at the lithospheric scale. The results of the models are impressive, and provide a very detailed framework for the interpretation of the tectonic processes observed on the field. The presentation of the results is very clear and will be useful for a wide audience, from undergraduate students to senior researchers. The impact of this study could even be wider than people working on ophiolites, as the processes of nappe exhumation modeled in this paper also occur in the most emblematic mountain belts such as the Himalaya or the European Alps. This paper indeed demonstrates (in the 'key parameters...' section) how nappe exhumation may occur without involving tectonic-climate interactions (as required in analogue experiments to reproduce the same pattern of nappe stacking and exhumation).

I therefore recommend publication of this paper as it is, as I think it is the most valuable work I have read so far in Nature communications. Congratulations to the authors for this very inspiring work.

We thank the reviewer for the encouraging comments. We think that the implemented changes following the comments of reviewers 1 and 2 further improved the applicability of our work for the geology and geodynamics communities.

References

- Maffione, M., Thieulot, C., Van Hinsbergen, D. J., Morris, A., Plümpner, O., and Spakman, W., 2015, Dynamics of intraoceanic subduction initiation: 1. Oceanic detachment fault inversion and the formation of supra-subduction zone ophiolites: *Geochemistry, Geophysics, Geosystems*, v. 16, no. 6, p. 1753-1770.
- Moulas, E., Burg, J.-P., and Podladchikov, Y., 2014, Stress field associated with elliptical inclusions in a deforming matrix: mathematical model and implications for tectonic overpressure in the lithosphere: *Tectonophysics*, v. 631, p. 37-49.
- Searle, M., and Cox, J., 1999, Tectonic setting, origin, and obduction of the Oman ophiolite: *Geological Society of America Bulletin*, v. 111, no. 1, p. 104-122.
- Searle, M. P., 2007, Structural geometry, style and timing of deformation in the Hawasina Window, Al Jabal al Akhdar and Saih Hatat culminations, Oman Mountains: *GeoArabia*, v. 12, no. 2, p. 99-130.

Reviewers' comments:

Reviewer #1 (Remarks to the Author):

After re-reviewing the manuscript by Porkalob et al., Unfortunately I was not satisfied with the revisions or the rebuttal to my earlier comments about the obduction history of ophiolite belts. The authors still neglect hard field data, petrological observations, P-T and U-Pb constraints in their simulation which are often in disagreement with their key result. The key observation the authors neglect is that HP rocks are only in specific locations under ophiolite thrust sheets and do not form extruded domes that neck the ophiolite. The phenomenon of HP-LT rocks under ophiolites is not universally observed and there is misunderstanding of the observations in Oman and in other locations about the distribution of HP rocks under ophiolites. In most cases, ophiolites with their inverted metamorphic soles, are underlain by unmetamorphosed pelagic-continental margin lithologies that do not show any evidence for being buried to sufficient depths or heated to more than 200 C. For example in Greece the Pindos and Vardar zone ophiolites have been emplaced above unmetamorphosed rocks and are separated by a domain of relatively unmetamorphosed rocks (the Pelagonian Zone) and in Oman the continental margin sediments beneath the ophiolite and its sole form a thin skinned thrust belt with no observed metamorphic minerals (not even chlorite-biotite grade). The HP rocks in Oman are restricted to Saih Hataf region and are structurally lower, and occur ~15 m.y. after the ophiolite sheet was emplaced across the continental margin. The entire thrust package has been affected by further tectonic deformation, folding and erosion.

This is unfortunate but the hard field and petrological observations from a vast quantity of work (also not properly cited) do not support the model the authors propose of far travelled ophiolite thrust sheets being explained by extrusion of HP continental margin rocks. If anything it is the other way around, ophiolite emplacement was followed by continental margin subduction and HP metamorphism, but this has been well-known for 20 years or more.

With these points in mind I unfortunately cannot support publication of the manuscript. Some further comments below about the problem I have with the idea the authors propose to explain 'far travelled ophiolite thrust sheets'.

The authors attempt to address a long standing 'mechanical problem' associated with ophiolite emplacement, namely how can a higher density fragment of oceanic lithosphere be emplaced onto the lower density continent? This has been stated for over 50 years. I agree, this is physically difficult to envisage, however as I stated in my previous review, there is a wealth of data and observations to suggest the ophiolite with the attached metamorphic sole was emplaced onto unmetamorphosed continental margin rocks and thrust over 150 km in Oman and other ophiolite belts.

The problem I see with ophiolite obduction is that we need to get an ophiolite (a fragment of oceanic lithosphere) to overlie lower density continental crust. As the authors state, this can work by one of two ways associated with slightly different reference frames: i) Either placing the high density ophiolite sheet onto the continent or ii) The continent subducts (goes under the ophiolite) (which remains stationary) and then buoyantly extrudes doming and thinning the overriding ophiolite. There are some specific tests we can do to straight away rule out each scenario.

1. What is the pressure (burial depth) of the vast majority of rocks under the ophiolite thrust sheet? If these rocks are all HP metamorphic rocks, then they reached a depth associated with a considerable overburden and most likely were all subducted and buoyantly extruded. However, if most of the rocks do not display HP metamorphism then the overburden was likely <15km thick (equating to 0.4 GPa; the likely thickness of the ophiolite thrust sheet).

- Because a large proportion of the rocks under ophiolites are not metamorphosed it requires the overburden of the ophiolite was not massively thick, maybe 15 km maximum. As I previously stated, the authors are wildly overestimating the volume of rock that reached high pressure metamorphic conditions under ophiolite belts. In their revised simulations, the authors still have large quantities of rocks that they call 'non metamorphic', however upon reviewing the simulation

snapshots, these rocks still reached in some cases 50 km depths or more during subduction. This is incorrect, as at 50 km depth we would expect at least blueschist facies conditions in a subduction environment. Furthermore, in the authors simulation they are predicting all the continental crust under ophiolite belts to have experienced HP metamorphic conditions. We know this is not the case from seismic profiles across the Oman mountains where the continental margin sediments are just thin skinned thrust sheets and overlay Arabian continental basement which did not reach high pressure conditions.

2. What are the ages of metamorphism and deformation in the sub ophiolitic rocks? If the continental margin was subducted and extruded beneath the stationary ophiolite (as the authors suggest) then we would expect a continual array of metamorphic ages associated with significant burial of oceanic-continental margin units during subduction in the rocks underlying the ophiolite. On the other hand, if the ophiolite was already being emplaced (thrust) onto the continental margin there would be limited metamorphism in the underlying rocks (same argument I make above; i.e. Haybi, Hawasina units in Oman) and therefore there would only be 2 metamorphic 'events' recorded: i) metamorphic sole formation associated with intraoceanic subduction, and ii) one HP-LT metamorphic episode related to the subduction of the continental margin some 15 Myrs later, once the ophiolite was already emplaced. This has already been very well documented by numerous authors.

- In Oman we only see two distinct metamorphic events in the underlying rocks (sole metamorphism in oceanic crust at 0.8-1.2 GPa and 700-850 C, and HP-LT eclogite blueschist metamorphism in As Sifah at 2.0-2.3 GPa, 550 C that are separated by 18 Myrs, this is also the case for many other ophiolite belts where some HP rocks are located beneath ophiolites. This suggests there are two distinct metamorphic episodes/processes and you need to exhume the metamorphic sole rocks before burying the HP-LT that represent the leading edge of the continental margin. If the sole rocks and (now underlying non-metamorphic rocks) were not exhumed we may see several or more prolonged metamorphic events in these rocks as they would have to be kept down there for that long. This is certainly not the case.

3. What is the spatial distribution of HP rocks under the ophiolites? If the ophiolite remained stationary as the authors suggest, then the extruded HP-LT metamorphic rocks would be expected to be regionally distributed under the ophiolite thrust sheet; this is clearly not the case. In Oman HP-LT rocks only occur in the east near or close to the locus of the paleo-subduction zone namely the As Sifah region and the Ruwi melange. No-where else in Oman do HP-LT rocks of any kind (>5 kbar) occur under the ophiolite which is at odds with the model results where even 'non metamorphic rocks still reached substantial pressures (>1.0 GPa)

The Title does not make any sense, the extrusion of the HP rocks is a consequence of ophiolite obduction and occurred 15 m.y. later than the ophiolite and sole were formed. It does not "explain" ophiolite obduction temporally or spatially. The Abstract says nothing new at all. We have known for many years the details of ophiolite obduction by looking at the sole rocks and details of the later subduction of the continental margin by studying the HP rocks. The big conclusion that the two processes are linked is absolutely not new. What new data do authors bring to this study? Simulating what many others have already said is not satisfactory. There is a wealth of detailed structural mapping, thermobarometry and U-Pb age data, all of which constrain the real tectonic evolution and almost none of these are referenced or used in this model (actually a very poor simulation).

Line 22. What is roof thrusting? I presume they mean passive roof thrusting – this has been widely proposed for extrusion of HP rocks for >15 years by numerous authors.

Line 33, Some metamorphic soles have pressures up to 10-12 kbar in the granulites, so cannot be described as LP.

Line 37. The HP rocks in Oman are not upper crust, they are likely middle crust, Lower Permian and Ordovician.

Fig. 1. Ophiolite thrust sheet width is purely down to erosion, or later folding structures, so a meaningless dimension when trying to explain obduction. In Oman the ophiolite width ranges from 0 – 150 km. The Subduction-exhumation cycle (myr) parameter, what is this referring to?- the metamorphic sole or the HP rocks? This diagram is not useful.

Fig. 2 is just imagination, and just plain wrong in many respects! There is no 'initial continental subduction' – where people have done a lot of PT and U-Pb dating work it shows that sole was all intra-oceanic subduction, and the continental margin only went down 15 m.y. after initiation of the subduction zone during the obduction, and after the far-travelled ophiolite thrust sheet was emplaced. The parameters are all made up, we have no idea of the dip of the subduction zone, no idea of the depth of the isotherms, no idea of the rate of subduction. Between (a) and (b) there is an entire obduction history missing: shortening of an ocean >150 wide, from sole formation to emplacement of the thin skinned thrust sheets. In Oman the 'far-travelled ophiolite sheet' was already emplaced at least 10 m.y. prior to subduction of the HP rocks. Why don't the authors use real field, metamorphic and timing constraints instead of just making these parameters up? Numerous references of key papers that do have real data are not quoted or used.

Line 119. The upper crust did not subduct to form HP rocks in Oman; the HP rocks exposed are middle crust (Permian-Ordovician) and only occur in a limited geographic location.

Figs 3 and 4 are just pure fantasy; there are no geological constraints put into this simulation, not a model. Many earlier workers have made detailed structural maps, constrained the depths using precise PT thermobarometry, and dating each phase of metamorphism. None of it goes into this model, or is even referenced.

Line 230. If shear heating was 'essential for nappe formation' why is there no metamorphic evidence of this at all in the unmetamorphosed rocks, there isn't even chlorite-biotite metamorphism in the thrust sheets immediately beneath the ophiolite.

Fig. 6. The basal thrust in this figure is labelled a 'roof thrust' which is incorrect. There are far better, more precise structural cross-section published than this one.

As said in my previous review, the metamorphic soles provide almost all the evidence for subduction and emplacement of ophiolites in the oceanic domain, these are almost entirely ignored in this paper. If the key point of this paper is relating far travelled ophiolites to HP subduction, the authors must describe and provide the data for the sole rocks. There is a large amount of data published on this in terms of mapping, structure, PT and age constraints.

I have seen the reply to reviewers' comments reveals a lack of any new data, and an imprecise interpretation of well-known terminology. You cannot have obduction without subduction. The ophiolite emplacement is almost certainly over and done with by the time the continental margin is subducted (see the well constrained age dating in Oman for example). What the authors call 'unmetamorphosed rocks' are still buried to significant depths. The fact that the ophiolite is emplaced over unmetamorphosed sedimentary rock is not from their model, it has been very well known for >40 years (this supports the idea that the ophiolite must already be emplaced onto the continent prior to HP-LT metamorphism). The changes to their model now are simply simulating what we already know.

The reply to Reviewers Comments are unsatisfactory. The reply to my comment 3 is not acceptable, the authors accept it and say it is their best estimate of the subduction cycle – It is not 'their estimate' - this has been proposed by many workers with proper data (structure, PT, U-Pb etc) none of which is cited. The authors still do not address the metamorphic sole emplacement history and this is key if they want to link ophiolite obduction to HP subduction. Anyway, this has been addressed before by numerous other authors with new structure, PT and age dating.

Overall I think the paper still needs some very careful rethinking considering in particular the distribution of HP metamorphic rocks under ophiolites belts, as stated many times in my review, there is a wealth of data that suggest the ophiolite are emplaced onto the continental margins prior to any HP-LT metamorphism and subsequent extrusion processes. The referencing is

extremely poor, the model is just a simulation, the conclusions are at odds with all the published geochronology and the interpretation/ conclusion is poorly supported by hard field/petrological/ /U-Pb data.

Thomas Lamont 29/10/2020

Reviewer #2 (Remarks to the Author):

I find the revised ms a real improvement on the original, both in the style of presentation, and in the redrafting and reorganising of the figures. I think the work on the figures, combined with the authors' improvements following the criticisms of referee #1, has made the whole story much easier to follow.

AS I was reading, I did some minor edits on the grammar etc, which are tracked on the attached version of the ms.

I recommend publication of thee revised ms

WL Griffin

Reviewer comments and replies for manuscript NCOMMS-20-28739

Below we provide our response to the comments of the reviewers and detail the modifications made to improve our manuscript. When referring to a figure, we generally mean the figure in the new revised version of the manuscript. For easier reading, the reviewer's comments are highlighted with blue.

Reviewers' comments:

Reviewer #1

Major Remark 1.

After re-reviewing the manuscript by Porkalob et al., Unfortunately I was not satisfied with the revisions or the rebuttal to my earlier comments about the obduction history of ophiolite belts. The authors still neglect hard field data, petrological observations, P-T and U-Pb constraints in their simulation which are often in disagreement with their key result. The key observation the authors neglect is that HP rocks are only in specific locations under ophiolite thrust sheets and do not form extruded domes that neck the ophiolite. The phenomenon of HP-LT rocks under ophiolites is not universally observed and there is misunderstanding of the observations in Oman and in other locations about the distribution of HP rocks under ophiolites. In most cases, ophiolites with their inverted metamorphic soles, are underlain by unmetamorphosed pelagic-continental margin lithologies that do not show any evidence for being buried to sufficient depths or heated to more than 200 C. For example in Greece the Pindos and Vardar zone ophiolites have been emplaced above unmetamorphosed rocks and are separated by a domain of relatively unmetamorphosed rocks (the Pelagonian Zone) and in Oman the continental margin sediments beneath the ophiolite and its sole form a thin skinned thrust belt with no observed metamorphic minerals (not even chlorite-biotite grade). The HP rocks in Oman are restricted to Saih Hatat region and are structurally lower, and occur ~15 m.y. after the ophiolite sheet was emplaced across the continental margin. The entire thrust package has been affected by further tectonic deformation, folding and erosion. This is unfortunate but the hard field and petrological observations from a vast quantity of work (also not properly cited) do not support the model the authors propose of far travelled ophiolite thrust sheets being explained by extrusion of HP continental margin rocks. If anything it is the other way around, ophiolite emplacement was followed by continental margin subduction and HP metamorphism, but this has been well-known for 20 years or more. With these points in mind I unfortunately cannot support publication of the manuscript. Some further comments below about the problem I have with the idea the authors propose to explain 'far travelled ophiolite thrust sheets'.

Firstly, the reviewer rejects the well-accepted, long-standing definition of obduction, i.e. the emplacement of oceanic lithosphere over a **continent** (e.g. Coleman, 1971; Dewey, 1976; Oxburgh, 1972). The definition implies that **intra-oceanic subduction** (where metamorphic soles form) **is not obduction**, because there is no continent involved in the process. It follows that an ophiolite sheet **cannot be emplaced prior to the subduction of the underlying continent**. Consequently, the claim of the reviewer: "*If anything it is the other way around, ophiolite emplacement was followed by continental margin subduction and HP metamorphism*" **is inconsistent with the definition of "obduction"**. In fact, most of the reviewer's comments are based on his view on how ophiolite emplacement works, which is at variance with the well-accepted definition as explained above. After the first round of review, we

already emphasized this point with the aim of making the reviewer realize that our model is consistent with the data describing ophiolite emplacement. As such, the often cited “hard geological observations” **do support** our model. The reviewer’s erroneous view that oceanic subduction equals obduction is a recurring comment in this document, and hence we will refer back to this argument on numerous occasions.

One major group of data specifically mentioned by the reviewer are radiometric ages derived from the metamorphic soles of the ophiolite sheets, formed during intra-oceanic subduction. For an unknown reason, the reviewer is under the impression that since we do not focus on the modelling of intra-oceanic subduction, our results are necessarily at odds with the data derived from the metamorphic soles. Below we will show that this conclusion is unjustified as demonstrated on the case of the Oman example, following the reviewer’s preference.

The available geological data indicates that 1) metamorphic sole formed during intra-oceanic subduction initiation occurred at 92-95 Ma; 2) Continental subduction of the passive margin initiated at ca. 87-88 Ma; 3) Peak HP-LT metamorphism of the subducted crust occurred at ca. 80 Ma; 4) Exhumation of the subducted crust to near surface/surface conditions occurred between 75 and 70 Ma (Agard et al., 2010; Searle et al., 2004; Searle, 2007 and references therein). Thus, hard geological data suggests that the whole tectonic evolution from intra-oceanic subduction to the final exhumation of the subducted continental crust occurred within 20-25 Myrs. In our model, this process takes 25 Myrs (Fig. 2), and hence fits very well to the data. However, the duration of the above sketched evolution depends greatly on the size of the oceanic domain, which is subducted prior to obduction. Therefore, it is critical to obtain a good fit regarding the timing of the **continental** subduction-exhumation cycle, the process that we study in the paper. In the case of Oman, the data suggests that the continental subduction-exhumation cycle took ca. 15 Myrs (see Fig. 1 and Supplementary Information for other natural examples). In our model, this is ca. 17 Myrs, and hence the fit is quite good with the hard geological data (see Fig. 1 for more examples). Thus, **the statement of the reviewer that our model does not fit hard geological data in Oman is wrong.**

A second major group of data that the reviewer keeps referring to is the metamorphic grade of the sub-ophiolitic rocks. We have already provided extensive explanations as well as an additional figure in the previous revised manuscript (Fig. 6 in the previous manuscript file) to show that our model is consistent with non-metamorphic and low-grade rocks below the ophiolites. We provide more details on this point below (Major Remark 3).

Major Remark 2.

The authors attempt to address a long standing ‘mechanical problem’ associated with ophiolite emplacement, namely how can a higher density fragment of oceanic lithosphere be emplaced onto the lower density continent? This has been stated for over 50 years. I agree, this is physically difficult to envisage, however as I stated in my previous review, there is a wealth of data and observations to suggest the ophiolite with the attached metamorphic sole was emplaced onto un-metamorphosed continental margin rocks and thrust over 150 km in Oman and other ophiolite belts.

Indeed, there is a wealth of geological data available from ophiolite belts. **We used these data and established a dataset of key criteria from 10 different ophiolite belts on Earth** (Fig. 1). Furthermore, we use these data for comparison and model validation. We agree that fundamental concepts of ophiolite

obduction have already been postulated based on geological observations providing a predominantly kinematic framework. However, improving our understanding of ophiolite emplacement requires a physics-based approach (through analogue or numerical modelling) to appreciate the dynamics of the system. Discarding modelling studies because there is also available geological data that constrain the same process demonstrates a lack of understanding of the added value of modelling studies. The aim of our study is to understand the physical conditions and evolution of continental subduction-exhumation and simultaneous emplacement of ophiolite sheets. Understanding the dynamics of such a complicated system only based on geological data is not possible, therefore physics based models are imperative in our view. Despite the wealth of data available from ophiolite belts, the link between the subduction and exhumation of continental rocks and far-travelled ophiolite emplacement is far from being clear as is the geodynamic forcing that explain the superposition of contraction and extension in ophiolite belts. Our models provide a dynamic, physics-based explanation for both these aspects, and therefore provide relevant, novel insights regarding the evolution of ophiolite belts. Regarding the last sentence of this comment '*emplaced onto un-metamorphosed continental margin rocks and thrust over 150 km in Oman and other ophiolite belts*', we provide our response below where this topic is discussed extensively (Major Remark 3).

Major Remark 3.

The problem I see with ophiolite obduction is that we need to get an ophiolite (a fragment of oceanic lithosphere) to overlie lower density continental crust. As the authors state, this can work by one of two ways associated with slightly different reference frames: i) Either placing the high density ophiolite sheet onto the continent or ii) The continent subducts (goes under the ophiolite) (which remains stationary) and then buoyantly extrudes doming and thinning the overriding ophiolite. There are some specific tests we can do to straight away rule out each scenario.

a) What is the pressure (burial depth) of the vast majority of rocks under the ophiolite thrust sheet? If these rocks are all HP metamorphic rocks, then they reached a depth associated with a considerable overburden and most likely were all subducted and buoyantly extruded. However, if most of the rocks do not display HP metamorphism then the overburden was likely <15km thick (equating to 0.4 GPa; the likely thickness of the ophiolite thrust sheet).

- Because a large proportion of the rocks under ophiolites are not metamorphosed it requires the overburden of the ophiolite was not massively thick, maybe 15 km maximum. As I previously stated, the authors are wildly overestimating the volume of rock that reached high pressure metamorphic conditions under ophiolite belts. In their revised simulations, the authors still have large quantities of rocks that they call 'non metamorphic', however upon reviewing the simulation snapshots, these rocks still reached in some cases 50 km depths or more during subduction. This is incorrect, as at 50 km depth we would expect at least blueschist facies conditions in a subduction environment. Furthermore, in the authors simulation they are predicting all the continental crust under ophiolite belts to have experienced HP metamorphic conditions. We know this is not the case from seismic profiles across the Oman mountains where the continental margin sediments are just thin skinned thrust sheets and overlay Arabian continental basement which did not reach high pressure conditions.

The reviewer's estimation of burial depth of the shallow parts of the continental crust below the oceanic upper plate through inspection of "*the simulation snapshots*" is wrong. In Fig. 3a we show that during the onset of extrusion (i.e. the stage of maximum burial), the shallowest part of the buried continental

crust (which will constitute the non-metamorphosed to low-grade part of the continental tectonic window after extrusion) is ca. 19 km (ca. 0.5 GPa if translated to lithostatic pressure) and not “50 km or more” as suggested by the reviewer. Consequently, the model shows no disagreement the presence of non-metamorphosed to low-grade continental units underlying the ophiolites, and **our model certainly does not predict that all continental crust below the ophiolite experienced HP metamorphism**. Since ophiolite sheets and the accreted sedimentary cover units (i.e. the composite upper plate of obduction systems) are typically at least 15 km thick in nature, the burial depth of the shallow continental crust in our model (19 km) is in the range of the minimum value that must be reached by all natural continents during ophiolite emplacement. We already stressed this point in our previous reply and supported it with a new figure (Fig. 6 in the previous manuscript version). In addition, the reviewer is incorrect when referring to the Pelagonian zone, separating the Vardar and Pindos ophiolite zones in Greece, as non-metamorphosed unit (see in Major Remark 1.); the Pelagonian zone was metamorphosed during the obduction of the ophiolites, showing a transition from low-grade to eclogite facies conditions (Kilias et al., 2010; Mposkos and Perraki, 2001; Porkoláb et al., 2019; Schenker et al., 2014).

In the new version of the manuscript, we improved Fig. 6 by plotting exact metamorphic facies conditions based on the maximum P-T conditions reached by the upper crust through the model evolution. Fig. 6a shows the case of the reference model, which has an initial continental geothermal gradient ($T_{\text{MOHO}} = 510$ °C), while 6b shows a variant to the reference model with an initially colder continental setting ($T_{\text{MOHO}} = 375$ °C). Even in case of the ‘hotter’ model version (reference model, Fig. 6a), ca. half of the extruded upper crust that is exposed on the surface consists of rocks that did not reach 300 °C, and are therefore non-metamorphic to very low-grade. In case of the colder model (Fig. 6b), the non-metamorphic to low-grade facies conditions dominate the upper crustal tectonic window, while eclogite facies rocks are only found in much deeper positions. This latter case appears to be a better fit for natural sites like Oman, where significant fraction of the exposed crust in the tectonic window is non-metamorphic or low-grade (as also stated by the reviewer) (e.g. Searle et al., 2004). The reference model case on the other hand fits better with natural sites like New Caledonia, where the exposed continental crust is largely metamorphic, showing a transition from blueschist facies to eclogite facies P-T conditions (Brovarone and Agard, 2013; Potel et al., 2006). These differences between natural cases are the main reasons for comparing the general modelling results with a dataset derived from 10 different ophiolite belts (Fig. 1) instead of focusing on a one-to-one comparison with a single ophiolite belt. In the new version of the manuscript, we discuss these differences between natural cases as well as between the different model versions, and hence improve on the model-nature comparison compared to the previous manuscript version.

b) What are the ages of metamorphism and deformation in the sub ophiolitic rocks? If the continental margin was subducted and extruded beneath the stationary ophiolite (as the authors suggest) then we would expect a continual array of metamorphic ages associated with significant burial of oceanic-continental margin units during subduction in the rocks underlying the ophiolite. On the other hand, if the ophiolite was already being emplaced (thrust) onto the continental margin there would be limited metamorphism in the underlying rocks (same argument I make above; i.e. Haybi, Hawasina units in Oman) and therefore there would only be 2 metamorphic ‘events’ recorded: i) metamorphic sole formation associated with intraoceanic subduction, and ii) one HP-LT metamorphic episode related to

the subduction of the continental margin some 15 Myrs later, once the ophiolite was already emplaced. This has already been very well documented by numerous authors.

- In Oman we only see two distinct metamorphic events in the underlying rocks (sole metamorphism in oceanic crust at 0.8-1.2 GPa and 700-850 C, and HP-LT eclogite blueschist metamorphism in As Sifah at 2.0-2.3 GPa, 550 C that are separated by 18 Myrs, this is also the case for many other ophiolite belts where some HP rocks are located beneath ophiolites. This suggests there are two distinct metamorphic episodes/processes and you need to exhume the metamorphic sole rocks before burying the HP-LT that represent the leading edge of the continental margin. If the sole rocks and (now underlying non-metamorphic rocks) were not exhumed we may see several or more prolonged metamorphic events in these rocks as they would have to be kept down there for that long. This is certainly not the case.

It is a mystery why the reviewer would think that the ages and P-T data of metamorphic soles and the following HP metamorphism of the crust do not fit with our model. We have argued extensively in our previous reply (and also above in this document), that in fact these data fit our model perfectly, and we actually use a large portion of them for validating our model (Fig. 1, Supplementary Information). A quote from our previous reply:

“The cited literature and the related evolutionary model of Oman fits very well our interpretation. 1) intra-oceanic subduction at ca. 95 Ma with the formation of the metamorphic sole (which is not in our focus); 2) oceanic subduction and the accretion of the sedimentary cover units; 3) initiation of continental subduction (=obduction) at ca. 85 Ma and HP-LT metamorphism from ca. 80 Ma; 4) exhumation (extrusion) of the subducted crust and final emplacement of the far-travelled ophiolite sheet at ca. 70 Ma. From the initiation of continental subduction (ca. 85 Ma) to the exhumation (ca. 70 Ma), the time span is 15 Myr. This is our best estimate for the continental subduction-exhumation cycle in Oman, and this is the 15 Myr plotted on Fig 1d in the original manuscript. In our reference model, the same evolution from continental subduction initiation to the exhumation of the HP units takes ca. 17 Myr, which is closely comparable.”

We stress once again that the evolution of ophiolite belts undoubtedly include two tectono-metamorphic stages: 1) ophiolite sole formation during intra-oceanic subduction, and 2) HP-LT metamorphism of the subducted continental crust during obduction. This is perfectly consistent with our model prediction (see Fig. 2). We do not see, as claimed by the reviewer, where hard geological data are at odds with our results.

c) What is the spatial distribution of HP rocks under the ophiolites? If the ophiolite remained stationary as the authors suggest, then the extruded HP-LT metamorphic rocks would be expected to be regionally distributed under the ophiolite thrust sheet; this is clearly not the case. In Oman HP-LT rocks only occur in the east near or close to the locus of the paleo-subduction zone namely the As Sifah region and the Ruwi melange. No-where else in Oman do HP-LT rocks of any kind (>5 kbar) occur under the ophiolite which is at odds with the model results where even ‘non metamorphic rocks still reached substantial pressures (>1.0 GPa)

The final state of our reference model shows a transition from non-metamorphic to HP-LT blueschist-eclogite facies rocks in the tectonic window below the ophiolite, similar to many natural ophiolite belts (Fig. 6). Our model suggests that the sub-ophiolitic, non-metamorphic/low-grade continental crust is structurally involved in the extrusion process, despite not having reached great burial depths (Figs. 3, 6). In the case of Oman, a deep ramp-flat thrust geometry below the non-metamorphosed continental units is inferred based on the dome geometry (e.g. Searle, 2007). In this sense, the non-metamorphic continental units in the dome were also involved in the thrusting in Oman and hence contributed to the deformation of the overlying ophiolite sheet. Thus, the general structural and metamorphic characteristics of our model and natural cases (e.g. Oman) show a lot of similarity. We do acknowledge, that the exact proportion of non-metamorphic and metamorphic crust in the tectonic window shows some deviation compared to Oman (there is a somewhat higher proportion of metamorphosed material in the model than in case of Oman) in case of our reference model (Fig. 6a). On the other hand, the model with a colder initial continental geotherm (Mod 1 variant) shows very good fit with the proportions of non-metamorphic and metamorphic crustal domains in Oman (Fig. 6b). This shows, that modelling results that are not tailored to a single natural case should be compared to general datasets (such as the one on Fig. 1). The one-to-one comparison of a general numerical model with an exact natural site will always show some differences and expecting otherwise shows a lack of understanding what modelling is aimed for. Our study provides physical mechanisms for the relation between exhumation of continental crust and the deformation (necking and breaking) of ophiolite sheets. Such a model is not aimed for reproducing the exact published cross sections from Oman; it is for understanding first-order features and underlying mechanisms. This is the reason why we use geological data for validation from 10 different ophiolite belts (Fig. 1); to provide a more general (global) frame of reference and avoid one-to-one comparisons with single cases, which is rarely beneficial for the overall understanding of the process. We emphasize that tuning the models to a single case was never the intention of the study (although the new version of Fig. 6 implies that it could be achieved). The reviewer has diverted the whole revision process of the manuscript towards a debate about the details of the Oman site. As we stress in the discussion of the manuscript:

“The precise reproduction of smaller scale nappes (nappe thickness of several kilometers) which is often observed in case of continental subduction would require very high-resolution numerical modeling and built-in heterogeneities inside the upper crust to localize shear zones at multiple horizons. Our results also support that crustal decoupling and exhumation may take place with different timing and position in the subducted continent depending on the rheology of the continental crust. Variations in thermal or compositional properties thus might control the surface preservation (exhumation) or the subduction and recycling of different types of continental passive margins.”

We thus highlighted in the discussion section that: 1) the homogenous nature of the upper crust in the model means that smaller-scale nappe stacking is difficult to reproduce. Hence, the exact structural patterns and the distribution of nappes with different metamorphic grade may be different compared to natural ophiolite sites. This is a limitation of the model, but it does not mean that the general physical processes derived from the model are not meaningful. It simply means that the one-to-one comparison with published cross sections (which is exactly what the reviewer was doing) will (and should) always exhibit some differences. 2) We show that the rheology of the crust controls the timing and location of extrusion. Thus, different continental passive margins will show different patterns of extrusion, and thus

different proportions of metamorphic/non-metamorphic rocks in the exhumed tectonic window. Our reference model is a version that shows good general fit with the geological data presented on Fig. 1, however, certain details of exact natural sites may fit better with different model setups using slightly different rheological/thermal parameters (e.g. the Oman case fits better with a colder initial continental geotherm, see Fig 6b). Please also see the reworked '*Comparison to natural ophiolite belts*' section in our revised manuscript, where this issue is specifically elaborated.

In summary, our results can be used to understand the dynamics of continental subduction-exhumation processes in obduction settings, and the related deformation of the oceanic upper plate that leads to far-travelled ophiolite emplacement. Are the results valid for specific examples like Oman? Yes, they are. They fit the general structure, metamorphic characteristics and history, and clearly help to gain a better understanding on the physical processes behind the geological observations. Can our model (or any large-scale geodynamic model for that matter) be treated as an exact cross section through a specific natural example? Obviously not; such geodynamic models are not aiming to reproduce nature, they are aiming to constrain processes, the physical links between the processes, and thus help to understand how geo-systems work and provide explanation for geological data.

Major Remark 4.

The Title does not make any sense, the extrusion of the HP rocks is a consequence of ophiolite obduction and occurred 15 m.y. later than the ophiolite and sole were formed. It does not "explain" ophiolite obduction temporally or spatially. The Abstract says nothing new at all. We have known for many years the details of ophiolite obduction by looking at the sole rocks and details of the later subduction of the continental margin by studying the HP rocks. The big conclusion that the two processes are linked is absolutely not new. What new data do authors bring to this study? Simulating what many others have already said is not satisfactory. There is a wealth of detailed structural mapping, thermobarometry and U-Pb age data, all of which constrain the real tectonic evolution and almost none of these are referenced or used in this model (actually a very poor simulation).

As explained in our previous reply and above in this document, **intra-oceanic subduction is not obduction** (therefore, the formation of the metamorphic sole is not occurring during obduction). Furthermore, we do not re-define obduction; it has been defined for decades as the emplacement of oceanic lithosphere over **continents** (e.g. Agard et al., 2014; Coleman, 1971; Dewey, 1976; Oxburgh, 1972). We merely keep this definition. The reviewer furthermore contradicts himself by stating both that 1) Our results are not correct because hard geological data does not support them (which is incorrect as detailed above); and 2) We ran simulations of the same processes that others have already shown via field work and geological data. This means that our results are correct (i.e. hard geological data support them), but they do not bring anything new (also false, as detailed below). It is furthermore striking that the reviewer had changed his opinion regarding the novelty of our study. In his first round of comments, he wrote: "*The manuscript made a very interesting read and investigates a very novel idea about extrusion of high-pressure rocks causing necking and thinning of the overlying ophiolite thrust sheet during subduction termination*", while in the second round he denied any novelty in the study.

The novelty of the manuscript is the thermo-mechanical explanation for the final emplacement of the far-travelled ophiolite sheets. We show for the first time, the nappe formation and extrusion of the crust changes the stress field of a convergent region to local extension, and results in the necking and breaking of the oceanic upper plate. Asking the question: "*What new data do authors bring to this*

study?” in a numerical modelling study is furthermore showing that the reviewer does not understand the added value of numerical modelling. As already explained, our novel numerical models enable the physical understanding of obduction systems and allow to explain the wealth of geological data better than previously. We show and in-detail explain the physical processes from buoyancy-driven decoupling in the subducted crust, through nappe formation, to the breaking of the oceanic upper plate in an unprecedented manner, highlighting physical links that were not clear before. Our results furthermore show a clear fit with general characteristics of natural ophiolite belts (Figs. 1, 6, Supplementary Info).

Smaller comments

1. Line 22. What is roof thrusting? I presume they mean passive roof thrusting – this has been widely proposed for extrusion of HP rocks for >15 years by numerous authors.

Indeed, this case is not strictly roof-thrusting, as the continental domain in the footwall is not a thrust nappe. This is more precisely a large-scale flat-ramp-flat thrust geometry, and here we are referring to the upper flat segment. To be clear, we changed the ‘roof thrust’ expressions to ‘upper flat segment’ at all locations in the text. Regarding the sentence *“this has been widely proposed for extrusion of HP rocks for >15 years by numerous authors”*: we acknowledge that other authors described extrusion processes in various geodynamic settings before us. We stress once again that the novelty of our study is linking extrusion and HP crust exhumation to the formation of far-travelled ophiolite sheets.

2. Line 33, Some metamorphic soles have pressures up to 10-12 kbar in the granulites, so cannot be described as LP.

Point taken, a range of LP/MP was implemented accordingly.

3. Line 37. The HP rocks in Oman are not upper crust, they are likely middle crust, Lower Permian and Ordovician.

In a differentiation into upper and lower crust as applied in our model all these rocks belong to the upper crust.

4. Fig. 1. Ophiolite thrust sheet width is purely down to erosion, or later folding structures, so a meaningless dimension when trying to explain obduction. In Oman the ophiolite width ranges from 0 – 150 km. The Subduction-exhumation cycle (myr) parameter, what is this referring to?- the metamorphic sole or the HP rocks? This diagram is not useful.

Naturally, ophiolite sheets are exposed to erosion, which modifies their original width. However, the currently measurable average width values are still important parameters, as they provide an indication for the original limits. We demonstrate in our study, that the width of the far-travelled sheet is not

purely down to erosion and later deformation, but there is a pre-defined maximum width of the ophiolite sheet that is determined by the location of the upper plate necking (Fig. 3). This relationship has not been suggested before and adds to the discussion on the dimensions of ophiolite sheets. The plotted subduction-exhumation cycle is referring to the continental rocks (passive margin). This is explained in the figure caption as well as detailed in the supplementary information. These data are extremely useful, as they outline a characteristic time range for subduction and exhumation of the continental rocks, which means that the responsible physical processes act according to a general pattern. It is also an excellent metric to use for validating our model, which addresses exactly the question of continental subduction-exhumation processes. As such we disagree on the usefulness of the diagram and argue that it carries critical information for constraining continental subduction-exhumation processes.

5. Fig. 2 is just imagination, and just plain wrong in many respects! There is no 'initial continental subduction' – where people have done a lot of PT and U-Pb dating work it shows that sole was all intra-oceanic subduction, and the continental margin only went down 15 m.y. after initiation of the subduction zone during the obduction, and after the far-travelled ophiolite thrust sheet was emplaced. The parameters are all made up, we have no idea of the dip of the subduction zone, no idea of the depth of the isotherms, no idea of the rate of subduction. Between (a) and (b) there is an entire obduction history missing: shortening of an ocean >150 wide, from sole formation to emplacement of the thin skinned thrust sheets. In Oman the 'far-travelled ophiolite sheet' was already emplaced at least 10 m.y. prior to subduction of the HP rocks. Why don't the authors use real field, metamorphic and timing constraints instead of just making these parameters up? Numerous references of key papers that do have real data are not quoted or used.

Claiming that the output of a robust numerical simulation is "*imagination*", or "*plain wrong*" and the used experiment-based, robust, properly cited parameters are "*made-up*" is **factually wrong**. These claims imply that the reviewer is unfortunately **not familiar with the very basics of numerical modelling and rock physics experiments**. Our models are based on robust experimental parameters and valid physical-mathematical considerations, as it is explained in the manuscript. The remaining parts of the comment (e.g. repeating the fact that oceanic subduction in Oman initiated 15 Myrs before continental HP metamorphism – a fact that is perfectly consistent with our model) are repetitions that were already replied to above (e.g. see reply to Major Remark 1).

6. Line 119. The upper crust did not subduct to form HP rocks in Oman; the HP rocks exposed are middle crust (Permian-Ordovician) and only occur in a limited geographic location.

See comment 3.

7. Figs 3 and 4 are just pure fantasy; there are no geological constraints put into this simulation, not a model. Many earlier workers have made detailed structural maps, constrained the depths using precise PT thermobarometry, and dating each phase of metamorphism. None of it goes into this model, or is

even

referenced.

Quite the contrary is true to what the reviewer claims. Our models are not '*pure fantasy*', but are based on robust experimental parameters and valid physical-mathematical considerations, as it is explained in the manuscript. We furthermore did use a very substantial amount of references to field geology-based studies in the manuscript as well as in the supplementary information.

8. Line 230. If shear heating was 'essential for nappe formation' why is there no metamorphic evidence of this at all in the unmetamorphosed rocks, there isn't even chlorite-biotite metamorphism in the thrust sheets immediately beneath the ophiolite.

Shear heating is merely a kick-starter of strain localization (e.g. Thielmann and Kaus, 2012), and also not the only mechanism leading to strain localization in rocks. The elevated temperature due to shear heating is not recorded (or not detectable) in most cases (Schmalholz and Duretz, 2015). Furthermore, shear heating only becomes important at the scale of large nappes and shear zones (Kiss et al., 2019), which is not the case with the thin slivers of accreted sedimentary units located immediately below the ophiolites. The importance of shear heating in this specific case is demonstrated by the lack of strain localization in the subducted crust when shear heating is not implemented in the model calculations (Fig. 4c).

9. Fig. 6. The basal thrust in this figure is labelled a 'roof thrust' which is incorrect. There are far better, more precise structural cross-section published than this one.

As stated in the figure caption, this figure is the output of our reference model, and thus in no way it is a structural cross section through Oman or any other place. This model output allows to describe the general near-surface features produced by the model, and thus link geological observation to the modelled physical processes. Claiming that there are better published cross sections for an actual natural example than the output of a general numerical simulation has absolutely no meaning. Regarding the roof thrust: we accepted the comment and changed it to "upper flat thrust segment" based on another comment of the reviewer above (Comment 1). Furthermore, we improved Fig. 6 for the new version of the manuscript (see our response to Major Remark 3).

10. As said in my previous review, the metamorphic soles provide almost all the evidence for subduction and emplacement of ophiolites in the oceanic domain, these are almost entirely ignored in this paper. If the key point of this paper is relating far travelled ophiolites to HP subduction, the authors must describe and provide the data for the sole rocks. There is a large amount of data published on this in terms of mapping, structure, PT and age constraints.

As already argued several times: intra-oceanic subduction is not obduction (per definition, based on several decades of studies (e.g. Agard et al., 2014; Coleman, 1971; Dewey, 1976; Oxburgh, 1972)); and intra-oceanic subduction has no direct link to the extrusion of continental rocks, as it occurs prior to the

subduction of the continental crust. We are completely aware of all the data related to intra-oceanic subduction (and we have demonstrated that they fit with our model), but we are not writing a paper on intra-oceanic subduction. **The claims that the lack of data presented on intra-oceanic subduction would invalidate our model are therefore factually wrong.**

11. I have seen the reply to reviewers' comments reveals a lack of any new data, and an imprecise interpretation of well-known terminology. You cannot have obduction without subduction. The ophiolite emplacement is almost certainly over and done with by the time the continental margin is subducted (see the well constrained age dating in Oman for example). What the authors call 'unmetamorphosed rocks' are still buried to significant depths. The fact that the ophiolite is emplaced over unmetamorphosed sedimentary rock is not from their model, it has been very well known for >40 years (this supports the idea that the ophiolite must already be emplaced onto the continent prior to HP-LT metamorphism). The changes to their model now are simply simulating what we already know.

The majority of this comment is a repetition of different previous comments, already replied to above (Major Remarks 1, 3). Regarding the "lack of any new data": in our previous reply we already demonstrated that our model is consistent with the available data (e.g. geochronological and P-T data from metamorphic soles and HP continental rocks) from ophiolite belts, also cited by the reviewer. We furthermore provided a figure in the reply document specifically comparing the evolution of our reference model with the Oman example. For unclear reasons the reviewer chose to ignore our arguments and favored repeating the same comments once again. We stress that our model is supported by the mentioned datasets, and thus providing 'new data' in our current reply is not needed. **In summary: the hard geological data from natural ophiolite belts (geochronological data from metamorphic soles and the subducted continent, thermodynamic P-T data, structural data) support our model (see the 30 cited references in Tables 1 and 2 in the Supplementary Information file or further citations in the 'Comparison to natural ophiolite belts' section, which all refer to hard geological data that validate our results).**

12. The reply to Reviewers Comments are unsatisfactory. The reply to my comment 3 is not acceptable, the authors accept it and say it is their best estimate of the subduction cycle – It is not 'their estimate' - this has been proposed by many workers with proper data (structure, PT, U-Pb etc) none of which is cited. The authors still do not address the metamorphic sole emplacement history and this is key if they want to link ophiolite obduction to HP subduction. Anyway, this has been addressed before by numerous other authors with new structure, PT and age dating.

This comment is again largely repetition. The part "*it is their best estimate of the subduction cycle – It is not 'their estimate' - this has been proposed by many workers with proper data (structure, PT, U-Pb etc) none of which is cited*" is **factually wrong**, the data on which our estimation is based are all properly cited in the supplementary material.

13. Overall I think the paper still needs some very careful rethinking considering in particular the distribution of HP metamorphic rocks under ophiolites belts, as stated many times in my review, there is a wealth of data that suggest the ophiolite are emplaced onto the continental margins prior to any HP-

LT metamorphism and subsequent extrusion processes. The referencing is extremely poor, the model is just a simulation, the conclusions are at odds with all the published geochronology and the interpretation/ conclusion is poorly supported by hard field/petrological/ /U-Pb data.

Thomas Lamont 29/10/2020

Once again: the claims that our model is at odds with the mentioned datasets is **factually wrong**, as demonstrated above. Our model is in fact consistent with the geological datasets from numerous natural ophiolite belts. We are certain that our model predictions are meaningful and provide new insights in the dynamics of ophiolite emplacement.

Reviewer #2 (Remarks to the Author):

I find the revised ms a real improvement on the original, both in the style of presentation, and in the redrafting and reorganising of the figures. I think the work on the figures, combined with the authors' improvements following the criticisms of referee #1, has made the whole story much easier to follow.

AS I was reading, I did some minor edits on the grammar etc, which are tracked on the attached version of the ms.

I recommend publication of thee revised ms

WL Griffin

We thank the reviewer for the constructive feedback that helped to make our manuscript easier to follow. The minor edits regarding the grammar are implemented in the new version of the manuscript.

References

- Agard, P., Searle, M. P., Alsop, G. I., and Dubacq, B., 2010, Crustal stacking and expulsion tectonics during continental subduction: P-T deformation constraints from Oman: *Tectonics*, v. 29, no. 5.
- Agard, P., Zuo, X., Funicello, F., Bellahsen, N., Faccenna, C., and Savva, D., 2014, Obduction: Why, how and where. Clues from analog models: *Earth and Planetary Science Letters*, v. 393, p. 132-145.
- Brovarone, A. V., and Agard, P., 2013, True metamorphic isograds or tectonically sliced metamorphic sequence? New high-spatial resolution petrological data for the New Caledonia case study: *Contributions to Mineralogy and Petrology*, v. 166, no. 2, p. 451-469.
- Coleman, R., 1971, Plate tectonic emplacement of upper mantle peridotites along continental edges: *Journal of Geophysical Research*, v. 76, no. 5, p. 1212-1222.
- Dewey, J., 1976, Ophiolite obduction: *Tectonophysics*, v. 31, no. 1-2, p. 93-120.
- Kilias, A., Frisch, W., Avgerinas, A., Dunkl, I., Falalakis, G., and Gawlick, H.-J., 2010, Alpine architecture and kinematics of deformation of the northern Pelagonian nappe pile in the Hellenides.

- Kiss, D., Podladchikov, Y., Duretz, T., and Schmalholz, S. M., 2019, Spontaneous generation of ductile shear zones by thermal softening: Localization criterion, 1D to 3D modelling and application to the lithosphere: *Earth and Planetary Science Letters*, v. 519, p. 284-296.
- Mposkos, E., and Perraki, M., 2001, High pressure Alpine metamorphism of the Pelagonian allochthon in the Kastania area (Southern Vermion), Greece: *Bulletin of the Geological Society of Greece*, v. 34, no. 3, p. 939-947.
- Oxburgh, E., 1972, Flake tectonics and continental collision: *Nature*, v. 239, no. 5369, p. 202-204.
- Porkoláb, K., Willingshofer, E., Sokoutis, D., Creton, I., Kostopoulos, D., and Wijbrans, J., 2019, Cretaceous-Paleogene tectonics of the Pelagonian zone: inferences from Skopelos island (Greece): *Tectonics*.
- Potel, S., Mählmann, R. F., Stern, W., Mullis, J., and Frey, M., 2006, Very low-grade metamorphic evolution of pelitic rocks under high-pressure/low-temperature conditions, NW New Caledonia (SW Pacific): *Journal of Petrology*, v. 47, no. 5, p. 991-1015.
- Schenker, F. L., Burg, J. P., Kostopoulos, D., Moulas, E., Larionov, A., and Quadt, A., 2014, From Mesoproterozoic magmatism to collisional Cretaceous anatexis: Tectonomagmatic history of the Pelagonian Zone, Greece: *Tectonics*, v. 33, no. 8, p. 1552-1576.
- Schmalholz, S., and Duretz, T., 2015, Shear zone and nappe formation by thermal softening, related stress and temperature evolution, and application to the Alps: *Journal of Metamorphic Geology*, v. 33, no. 8, p. 887-908.
- Searle, M., Warren, C., Waters, D., and Parrish, R., 2004, Structural evolution, metamorphism and restoration of the Arabian continental margin, Saih Hatat region, Oman Mountains: *Journal of Structural Geology*, v. 26, no. 3, p. 451-473.
- Searle, M. P., 2007, Structural geometry, style and timing of deformation in the Hawasina Window, Al Jabal al Akhdar and Saih Hatat culminations, Oman Mountains: *GeoArabia*, v. 12, no. 2, p. 99-130.
- Thielmann, M., and Kaus, B. J., 2012, Shear heating induced lithospheric-scale localization: Does it result in subduction?: *Earth and Planetary Science Letters*, v. 359, p. 1-13.